

# Current status and grand challenges for small wind turbine technology

Alessandro Bianchini[1,17], Galih Bangga[2,17], Ian Baring-Gould[3], Alessandro Croce[4,17], José Ignacio Cruz[5], Rick Damiani[6], Gareth Erfort[7,17], Carlos Simao Ferreira[8,17], David Infield[9], Christian Navid Nayeri[10,17], George Pechlivanoglou[11], Mark Runacres[12,17], Gerard Schepers[13,14], Brent Summerville[3], David Wood[15], Alice Orrell[16]

[1] Department of Industrial Engineering, Università degli Studi di Firenze, Firenze, 50139, Italy
[2] Institute of Aerodynamics und Gas Dynamics, University of Stuttgart, Stuttgart, 70569, Germany
[3] National Renewable Energy Laboratory (NREL), Golden, Colorado, 80401, USA
[4] Dipartimento di Scienze e Tecnologie Aerospaziali, Politecnico di Milano, Milano, 20156, Italy
[5] Centro de Investigaciones Energéticas, Medioambientales y Tecnológicas (CIEMAT), Madrid, 28040, Spain
[6] RRD Engineering, LLC, Arvada, CO 80007, USA
[7] Department of Mechanical & Mechatronic Engineering, Stellenbosch University, Stellenbosch, 7602, South Africa
[8] Delft University of Technology, Wind Energy, 2629HS, Delft, The Netherlands
[9] University of Strathclyde, Glasgow, G1 1XW, Scotland
[10] Hermann-Föttinger-Institut, Technische Universität Berlin, Berlin, 10623, Germany
[11] Eunice Energy Group, Athens, 15125, Greece
[12] Vrije Universiteit Brussel, Brussels, 1050, Belgium
[13] TNO Energy Transition, Petten, 1755 LE, the Netherlands
[14] Hanze University of Applied Sciences, Groningen, 9747 AS, the Netherlands
[15] Department of Mechanical and Manufacturing Engineering, University of Calgary, T2N 1N4, Canada
[16] Pacific Northwest National Laboratory, Richland, WA 99352 USA
[17] Small Wind Turbine Technical Committee, European Academy of Wind Energy (EAWE)

*Correspondence to*: Alessandro Bianchini (alessandro.bianchini@unifi.it)

**Abstract.** While modern wind turbines have become by far the largest rotating machines on Earth with further upscaling planned for the future, a renewed interest in small wind turbines is fostering energy transition and smart grid development. Small machines have traditionally not received the same level of aerodynamic refinement of their larger counterparts, resulting in lower efficiency, lower capacity factors, and therefore a higher cost of energy. In an effort to reduce this gap, research programmes are developing worldwide. With this background, the scope of the present study is twofold. In the first part of this paper, an overview of the current status of the technology is presented in terms of technical maturity, diffusion, and cost. The second part of the study proposes five grand challenges that are thought to be key to fostering the development of small wind turbine technology in the near future, i.e.: (1) improve energy conversion of modern SWTs through better design and control, especially in the case of turbulent wind; (2) better predict long-term turbine performance with limited resource measurements and prove reliability; (3) improve the economic viability of small wind energy; (4) facilitate the contribution of SWTs to the energy demand and electrical system integration; (5) foster engagement, social acceptance, and deployment for global





distributed wind markets. To tackle these challenges, a series of unknowns and gaps are first identified and discussed. Based
on them, improvement areas are suggested within which ten key enabling actions are finally proposed.
**1 Introduction**
A major portion of today's installed wind power is in the form of large wind power plants, which mainly consist of multi-MW
machines (GWEC, 2020), while a clear trend in further upscaling of both rated power and dimension is ongoing (Veers et al.,
2019). Small wind turbines (SWTs) are, however, still visible around the world for a variety of applications, including electric
power generation for households, industrial centres, farms, and isolated communities; combining with other energy sources
and storage in hybrid energy systems for electricity to support remote monitoring and telecommunications; and providing
direct energy services for applications such as water pumping, desalination, and purification (Chagas et al., 2020). The use of
wind turbines in rural areas is of particular relevance for some countries; for example, around the horn of Africa, small wind
systems are the most viable solution in the scarcely electrified parts of those countries (Gabra et al., 2019). (Karekezi, 2002)
reported that South Africa has more than 100,000 wind pumps in operation used over 45,818 farms. SWTs are a subset of a
larger distributed wind market segment that can include large turbines installed in distributed applications. Figure 1 associates
typical distributed turbine sizes to their main types of application.

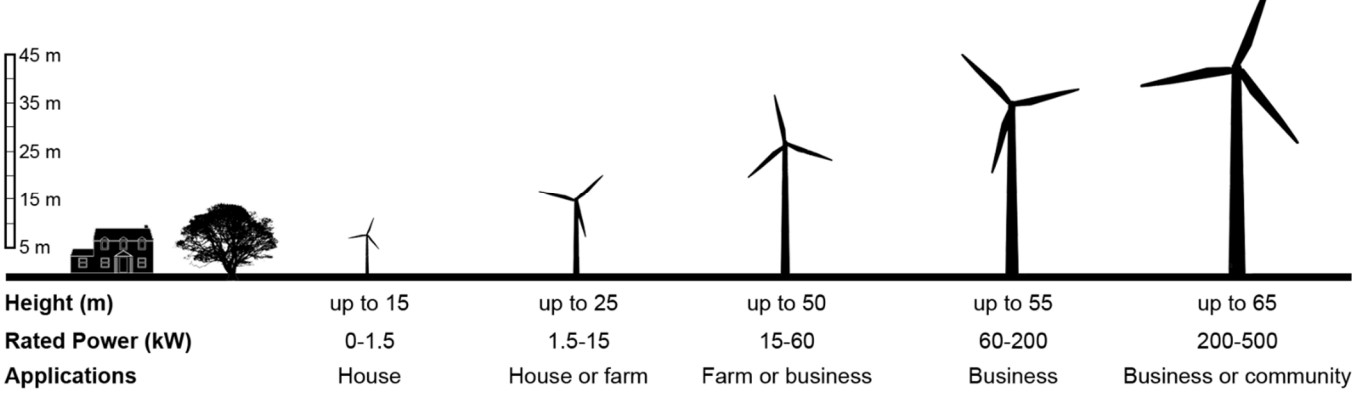

**Figure 1 - Small and distributed wind turbine dimensions and rated power outputs as a function of various applications.**
When SWTs are used for a variety of ancillary purposes other than electricity production such as ventilation or water pumping,
different turbine concepts can come to play. These applications may use the Savonius vertical-axis turbine (Akwa et al., 2012)
or the multi-blade American windmill (Baker, 1985), which each constitute a small space in the market. Although these
machines are in all respects SWTs, they are not discussed in the present study, which instead focuses on SWTs for electricity
production.



Before moving forward, a key element of this study is defining what is meant by "small wind turbine." A universal consensus
on this has not been reached, with the International Electrotechnical Commission (IEC) Standards (IEC: International Standard,
2019b) defining SWTs as turbines with a maximum rotor swept area of 200 m$^2$; the same threshold is applied to eligible
turbines for certification by the AWEA Small Wind Turbine Performance and Safety Standard 9.1-2009; however, a new
American National Standards Institute consensus standard, ACP 101-1, is being developed by the American Clean Power
Association (ACP), the successor to AWEA. ACP 101-1 is intended to eventually supersede the AWEA 9.1-2009 standard
(Summerville et al., 2021). Several countries use rated power as the key differentiator, and ACP 101-1 thus defines SWTs as
having a peak power of 150 kW or less and microturbines as having a peak power up to 1 kW. In Brazil, small wind systems
are categorized as power stations (which could be composed of one or many wind turbines) with a total rated capacity below
100 kW, according to Resolution 438/2012 of the Brazilian Electricity Regulatory Agency (ANEEL) (Chagas et al., 2020).
The importance of having a more comprehensive definition of "small wind" has been recently put in the spotlight. For example,
it has been suggested by the Small Wind Turbine Technical Committee of the European Academy of Wind Energy (EAWE)
that many problems and technical challenges of SWTs are common to the majority of the rotors up to 500 kW (EAWE, 2020),
i.e., also extending to distributed wind turbines (DWTs). As will be further discussed in the present study, it is important to
more clearly define those characteristics that make SWTs unique from utility-scale turbines. However, this is not an easy task
because significant variability in wind turbine design is also apparent, with no specific size-based design threshold.
Additionally, there are a variety of "alternative" configurations available on the open market (Bianchini, 2019), such as
vertical-axis turbines (Aslam Bhutta et al., 2012), diffuser augmented wind turbines (Evans et al., 2020), or first prototypes of
airborne wind energy (AWE) converters (Meghana et al., 2022). Even though SWTs may still represent a niche application
within the wind energy market, they have recently been exhibiting a notable rate of growth concomitant with the diffusion of
smart energy systems (Tzen, 2020). This diffusion, however, is still hindered by the typically higher costs of small wind
systems. These increased costs are driven by several factors, including a lack of development and system optimization and
issues related to those cost items (i.e., electrical connection, resource assessment expenses, installation cost, etc.) that are not
proportionally lower for smaller projects (Simic et al., 2013). The growth of the SWT sector is further notable in light of the
several published reports showing that SWT installations have failed to reach their expected energy yield, resulting in
underperforming turbines. This is particularly true in the case of installations in the urban or built environment (WINEUR
project, 2005; Fields et al., 2016). Development in highly complex areas, such as urban locations, is complicated due to the
wind conditions in the city's canopy layer, which typically have low intensity, high variability, high levels of turbulence, and
inclined or even reversed air flows. While several studies have shown a theoretically good potential for urban wind (Balduzzi
et al., 2012; Toja-Silva et al., 2013), a number of challenges still need to be tackled to effectively fit wind energy converters
to this environment, as recently discussed by (Micallef and Bussel, 2018) (Stathopoulos et al., 2018). In the present study, the
authors decided not to include urban wind specifically, although future work on the topic has to be encouraged (Battisti, 2018).
Even so, projections of SWT deployment in future scenarios of distributed energy production within smart grids (thus in
proximity to populated areas) are considered promising. In this sense, SWTs are expected to provide a significant contribution,



especially in combination with other renewable energy sources. However, the higher levelized cost of energy (LCOE) of SWTs, especially compared to residential solar photovoltaics (PV), still hampers the massive diffusion of this technology.

## 1.1 A guide to this article

The present study has two main focuses. First, it provides an overview on the status of SWT technology. We present the market diffusion and economics of SWTs (Sections 2–3) with the goal of placing the technology in the current energy market and defining some important threshold values. We then provide a description of the main technical features of SWTs (Section 4) and compare them to those of their utility-scale counterparts. Section 5 pursues the second focus of the work, defining five grand challenges that—per the authors' assessment—are key to fostering the development of SWTs in the near future. More specifically, a series of unknowns and gaps for SWTs is first defined, and then main improvement areas and prospects are proposed to address those gaps. Finally, Section 6 synthetizes the main outcomes of the study into concluding remarks and defines 10 key enabling actions for achieving the grand challenges in the near future.

## 2 Diffusion of small wind turbines

There is at least ~1.8 GW of installed small wind capacity globally from over 1 million turbines (Orrell et al., 2021). The global spread of this electrical capacity, based on available reports from some key surveyed countries, is shown in Table 1 (asterisks denote a lack of validated data for that specific year). Figure 2 provides a more focused insight into several of those countries, which showed notably different trends in the first years of the last decade, where SWT technology saw one of its more interesting phases. While Denmark, the United Kingdom, and the United States have a long-recorded history of small wind installations, China has added larger amounts of small wind capacity more consistently in recent years. On the other hand, Italy, and the United Kingdom, which saw many installations in the first decade of the century, both experienced recent decreases due to feed-in tariff (FIT) policy changes. FITs provide payments to owners of small-scale renewable generators at a fixed rate per unit of electricity produced, verifying that the cost of the installation is recovered over the lifetime of the generator. In the case of Italy, in particular, the significant increase in installations seen around 2016–2017 was due to a special programme of incentives for turbines under 60 kW. The FIT rate in Italy declined over time before expiring in 2017. It was replaced by the FER1 Decree in 2019 (Dentons, 2020). In line with these changes, an estimated 77.46 MW of wind projects using turbines sized up through 250 kW were installed in Italy in 2017, no installation reports were available for 2018 and 2019, and 0.65 MW of projects were reported for 2020. The United Kingdom closed its FIT programme to new applicants in 2019 and introduced the Smart Export Guarantee programme. Under that programme, applicants now receive a tariff determined by the buyer rather than a fixed price determined by the government (Ofgem, 2021). Consequently, small wind deployment went from 28.53 MW in 2014 to only 0.43 MW in 2019 (Orrell et al., 2021). In a scenario of decaying government incentives, an outlier case in Europe is Greece (Greek Government Gazette, 2021), which still offers an FIT for SWTs. At the time of writing this paper, the programme was for 20 MW installed capacity, starting with a tariff of 157 €/MWh (181 $/MWh)





that will be automatically reduced based on the cumulative contracted power of the projects. A bonus with respect to the tax
break is also in place, which brings the FIT to 163€/MWh (187 $/MWh).

**Table 1 - Small wind turbine installations through 2020. Data from (Orrell et al., 2021) and (Chagas et al., 2020).**

| | Installations (MW) | | | | | | | | | Cumulative (MW) installations | Cumulative Year Range |
|---|---|---|---|---|---|---|---|---|---|---|---|
| | Before 2012 | 2013 | 2014 | 2015 | 2016 | 2017 | 2018 | 2019 | 2020 | | |
| **Brazil** | 0.00 | 0.03 | 0.02 | 0.11 | 0.04 | 0.11 | 0.09 | * | * | 0.40 | 2013–2018 |
| **China** | 280.01 | 72.25 | 69.68 | 48.60 | 45.00 | 27.70 | 30.76 | 21.40 | 25.65 | 610.61 | 2007–2020 |
| **Germany** | 24.55 | 0.02 | 0.24 | 0.44 | 2.25 | 2.25 | 1.00 | * | * | 30.75 | As of 2018 |
| **Denmark** | * | * | * | * | 14.61 | 2.58 | 0.40 | 0.18 | 0.05 | 610.88 | 1977–2020 |
| **Italy** | 20.99 | 7.00 | 16.27 | 9.81 | 57.90 | 77.46 | * | * | 0.65 | 190.08 | As of 2018 |
| **South Korea** | 2.99 | 0.01 | 0.06 | 0.09 | 0.79 | 0.08 | 0.06 | * | * | 4.08 | As of 2018 |
| **United Kingdom** | 77.98 | 14.71 | 28.53 | 11.64 | 7.73 | 0.39 | 0.42 | 0.43 | * | 141.51 | As of 2019 |
| **United States** | 130.73 | 5.60 | 3.70 | 4.30 | 2.43 | 1.74 | 1.51 | 1.30 | 1.55 | 152.65 | 2003–2020 |
| **Other countries** | * | 1.65 | 1.32 | 6.23 | 5.40 | 3.39 | 13.23 | * | * | 33.72 | mixed ranges |
| | | | | | | | | **TOTAL** | | **1774.68** | |


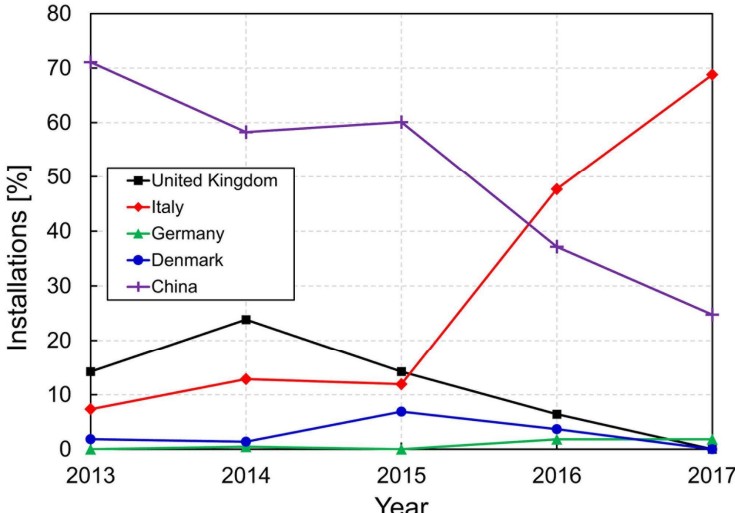


**Figure 2 - Evolution of the country's share in the newly installed SWT capacity for that year for a number of key European**
**countries and China. Data from (Orrell et al., 2021).**




Other examples of these tariffs include those in Japan and the Republic of Korea. Japan's FIT programme was established in
the wake of the Fukushima Daiichi nuclear disaster. Its rates have steadily declined, from a peak of ¥55 per kWh in 2015 to
¥19 (approximately[1] 0.125 € or 0.175 $) per kWh as of 2019 for turbines less than 20 kW (Orrell et al., 2021). The Republic
of Korea also had an FIT programme, but it was ended in 2012 and replaced with a renewable portfolio standard (Lo, 2018).
While the switch from the FIT programme increased capacities in some renewables in the Republic of Korea, such as biomass
co-firing and fuel cell deployment, small wind installations dropped (Orrell et al., 2021).
The discontinuous nature of incentives and national programmes makes it difficult for manufacturers to stay on the market,
even in those countries where SWT technology is more present, as in the UK, Italy, and the United States. Six small wind
manufacturers in the United States reported international exports in 2015, with just three doing so in 2020 (Orrell et al., 2021).
Similarly, sales in China and exports from China have fluctuated with the number of Chinese small wind manufacturers in that
market. In 2017, only 15 Chinese small wind turbine manufacturers reported sales, a decrease from 28 in 2014 (Duo, 2017),
corresponding to a 60% drop in sales from 2014 to 2017 (Orrell et al., 2021).
From a global perspective, at the time of writing this paper, the largest market for small wind still came from Europe, United
States, and China. SWTs are most commonly used for off-grid applications, such as telecommunication towers and farming.
They are also used to power individual homes and small businesses, which can be tied to the grid. In 2019, 94% of SWT sales
went to off-grid applications (Global Info Research, 2021). Unfortunately, 2020 saw only about 30 MW worth of units being
sold around the world (Orrell et al., 2021), with a global market in terms of revenues (Figure 3) still on a flat trend. Regarding
future perspectives (Global Info Research, 2021), no clear agreement on future perspectives was found at the time of writing,
mainly as a consequence of the financial crisis connected to the global COVID-19 pandemic in 2020. Global Info Research
(Global Info Research, 2021) predicted the SWT global market would reach 190 million USD (165 million EUR) in 2025 with
a compound annual growth rate of 11.45% from 2020 to 2025. The market could thus become promising again, especially in
connection with the increasing attention on the transition toward cleaner energy systems. Regarding the future share by region,
Europe, Asia-Pacific, and the United States are expected to remain the key players in this sector. In particular, the Asia-Pacific
market will lead the total worldwide SWT sales, while the European market will show a reduction in the global relative share
(Table 2). In Asia, Japan is expected to deploy renewable energy generation at large scales following the Fukushima Daiichi
nuclear disaster, whereas other countries such as Malaysia—which represents an untapped market with suitable conditions for
SWTs (Wen et al., 2019)—might also see significant deployment.

---

[1] Conversion rates used in the paper at the time of writing: 1¥ = 0.008€; 1€ = 1.15$ (USD).



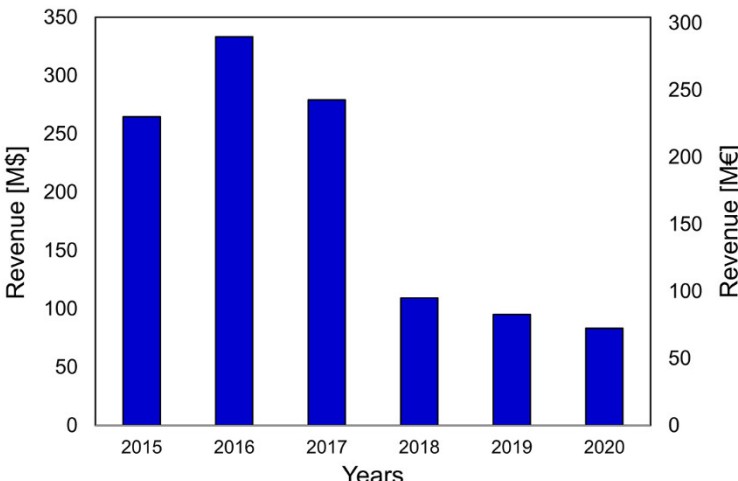

**Figure 3 - Global SWT market status in terms of revenues (Global Info Research, 2021).**

**Table 2 – Global SWT sales forecast by region (2020–2025). Data from (Global Info Research, 2021).**

| Sales (kW) | Forecast | | | | | |
|---|---|---|---|---|---|---|
| | **2020** | **2021** | **2022** | **2023** | **2024** | **2025** |
| **North America** | 1912 | 2185 | 2430 | 2736 | 3094 | 3414 |
| **Europe** | 4189 | 7015 | 6077 | 7033 | 8098 | 9084 |
| **Asia-Pacific** | 24993 | 29448 | 36575 | 41414 | 48960 | 58923 |
| **Others** | 873 | 962 | 1041 | 1182 | 1304 | 1454 |
| **Global** | 31967 | 39610 | 46123 | 52365 | 61456 | 72875 |

## 3 Economic aspects

As described in Section 2, the diffusion of SWTs has often gone hand-in-hand with dedicated financial incentive programmes from individual countries. This is unfortunately because the high LCOE of SWTs has represented the main obstacle hampering wider deployment of SWT technology (Predescu, 2016).

The economic evaluation of small wind systems is particularly critical for three main reasons: (1) the capital investment is strongly dependent on the specific turbine and country, (2) the correct selection of the installation site has a much higher impact on actual annual energy production (AEP) than in the case of turbines with large rotors, and (3) as discussed, the real viability of a project may depend completely on the incentives ensured by the specific country.

To give the reader an overview on the aforementioned issues, the main cost factors are analysed in the following subsections to facilitate the comparison of costs by country or region for the same technologies and to enable the identification of the key

 

drivers in any cost differences. The four key indicators are: total installation cost, operation and maintenance cost, capacity
factors, and LCOE.

### 3.1 Total installation cost

The total investment for installation can be expressed as the sum of the purchase cost and installation cost. The purchase cost
for an SWT is notably variable not only as a function of the turbine size but also over time, depending on the attention given
to the technology. (Kaldellis and Zafirakis, 2012) present a survey on 142 SWT models up to 20 kW, showing—as expected—
a turbine cost reduction as a function of the rated power (black square markers in Figure 4). Recent data from the authors'
direct experience are also added as red diamonds in Figure 4 for the SWTs with rated power outputs around 50 kW. As seen
in the figure, the decreasing cost trend for lower rated power values is somehow stopped or reversed when going over 50 kW.
This can be explained considering that, from this size up, turbines become more complex, requiring specific features (e.g., the
yawing system) and a manufacturing quality higher than that of smaller turbines. Finally, (Bortolini et al., 2014) provide a
more up-to-date market survey considering several producers located worldwide and confirm that purchasing costs are not so
highly correlated to the plant sizes because of aspects related to the specific producer, e.g., producer country, producer cost
structure, and market policies. Having direct information on how the global, or total installed, cost comes together is very rare.
In this study, thanks to support from Eunice Energy Group, a cost breakdown is presented in Table 3 for the 60 kW machine
EW16 Thesis (Eunice Energy Group, 2021).

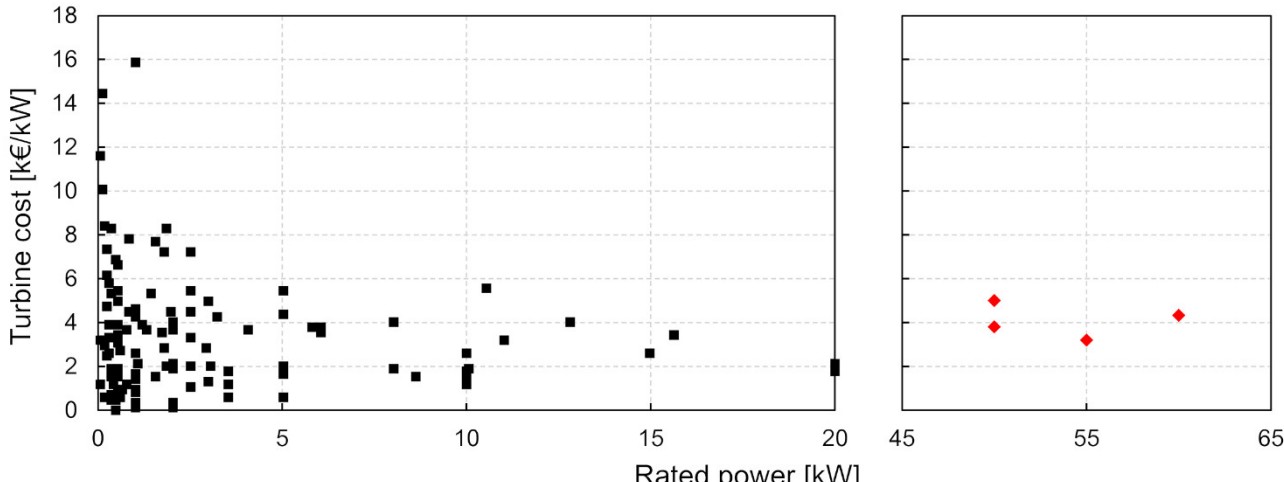

**Figure 4 - Turbine purchase cost survey for rated power lower than 20 kW (Kaldellis and Zafirakis, 2012) and around 50 kW**
**(authors' experience).**





**Table 3 – Capital cost breakdown of a 60 kW turbine (courtesy of Eunice Energy Group).**

|  | Cost | % of the total |
|---|---|---|
| **Tower** | ≈7 k€/ton (≈7 k$/ton) | 18% |
| **Generator** | ≈13 k€/ton (≈15 k$/ton) (permanent magnets) | 21% |
| **Gearbox (1:20)** | 8-10,000 € (9-11,500 $) | 5% |
| **AC-DC-AC converter** | 0.23 €/W (0.265 $/W) | 7% |
| **Blades** | 20 €/kg (23 $/kg) | 4% |
| **Rest of machinery** | 12 €/kg (14 $/kg) | 5% |
| **Rest of materials** | 13-15 €/kg (15-17 €/kg) | 15% |
| **Labour cost and standard industrial profit** | - | 25% |


(Wood, 2011) reported a similar breakdown for a smaller machine (10 kW), showing how—in that case—the relative cost for
blades becomes more relevant (7%), while that of the generator becomes less significant (6%) due to the lower power output.
The installation cost is probably the most critical parameter to evaluate and includes seven primary factors:
1)  Raw material cost, i.e., expenditures to purchase the materials required for the turbine installation as well as to lay

the foundation. All these elements are correlated to the wind turbine's weight and height and to the rotor diameter

2)  Earthworks' cost, i.e., foundations, grounding, etc. to enable SWT's operation. This is more crucial for countries with

higher seismic activity that require more expensive foundations and is dependent on the type of soil

3)  Installation labour cost, i.e., workers' salary, crane rental, stand-by times on windy days
4)  Engineering cost, i.e., expenditures for the preliminary and executive drawings, feasibility study and engineering, and

site assessment and wind resource assessment activities to estimate expected AEP; documentation of all deliverables

5)  Land purchase cost, i.e., cost for the required ground surface. Considering the tower height, a surface area of the same

swept radius is assumed to be necessary. Additional cost for access roads, where not present, may be necessary

6)  Grid connection cost, i.e., cables, power unit, and control system, including licence fees
7)  Transportation costs, i.e., the expenditures necessary to get the turbine to the installation site. Transportation costs

can include two different types of trips. In the case of imported turbines, both transportation by sea (e.g., to reach the

EU mainland) and by land (i.e., to reach the final site) are needed.

The relative impact of these factors has been quantified by (Bortolini et al., 2014) and reported in Table 4.

**Table 4 – Impact of different cost factors on an SWT project.**

| Cost Factor | Impact [% of Global Cost] |
|---|---|
| **Purchase** | 76% |
| **Building material** | 7% |



| Labor | 2% |
|---|---|
| Engineering | 1% |
| Land purchase | 10% |
| Grid connection | 2% |
| Transportation | 2% |



Referring again to the 60 kW EW16 Thetis machine by Eunice Energy Group, even though real costs are strictly project-
dependent, the foundation cost can be broken down into approximately 3,000 € (3,450 $) for the excavation (23%), 8,000 €
(9,200 $) for the concrete (61%), and 2,000 € (2,300 $) for civil works (16%). The transportation cost is approximately 5,000
€/day (5,750 $/day) (up to two trucks, and up to 600 km), while the crane costs for a 50 t, 40 m crane are about 6,000 €
(7,200 $).
An overview of the overall average annual and project-specific small-wind installed cost (in 2020 USD) in the United States
for 2010 through 2020 is presented in Figure 5 (data from Orrell et al., 2021). Only new and retrofit projects with reported
installed costs that use turbines with known rated capacities are included. Annual average capacity-weighted installed costs
for new U.S. small wind projects range from around $4,000/kW (3,480 €/kW) to nearly $11,000/kW (9,565 €/kW). The small
sample sizes and high variance in project-specific costs both contribute to this wide cost range. With the exception of 2018,
the overall annual average capacity-weighted installed cost for this U.S. dataset has remained relatively flat at approximately
$9,500/kW (9260 €/kW) (Orrell et al., 2021). This cost trend is in contrast with residential solar PV costs, which have been
steadily dropping over several years (Barbose and Darghouth, 2015).

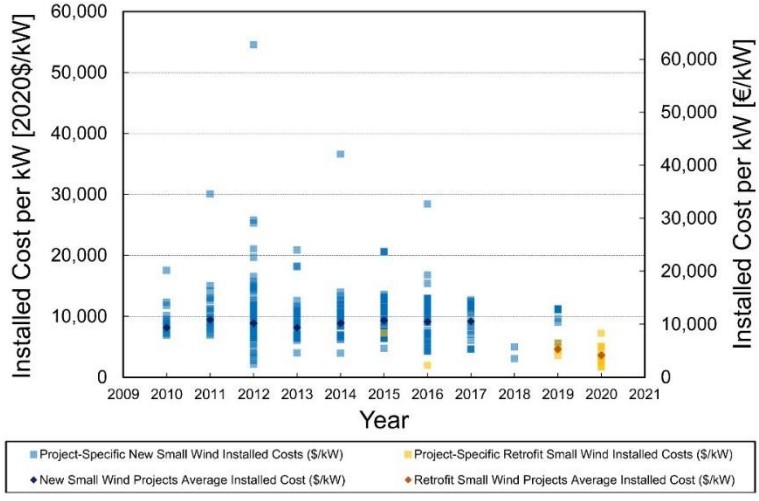


**Figure 5 - Installed cost per kW for new installed or retrofit installed projects in the United States (Orrell et al. 2021).**



## 3.2 Operations and maintenance cost

Operations and maintenance (O&M) are conventionally clustered into a single cost term, but operation costs differ from maintenance costs, and not all distributed wind projects experience them equally. Operation costs for wind projects may include land lease payments, remote monitoring, various operations contracts, insurance, and property taxes. Operations are a significant expense for wind farms and large distributed wind projects; however, they typically are not substantial, or even present, for small, distributed wind projects. On the other hand, all wind projects, distributed or otherwise, require a significant maintenance cost (Orrell et al., 2021). For small wind systems, and especially in the case of complex areas, experience shows that usually an investor does not opt for installation sites with more than two SWTs in the same field/owner. This consequently decreases the available room for the economy scaling on the O&M costs.

In most cases, the project installer or developer performs the maintenance for the system owner. Maintenance costs include labour, travel to the site, consumables, and any other related costs. Therefore, small wind maintenance costs can depend on the maintenance provider's proximity to the project site (i.e., travel costs), the availability of spare parts, and the complexity of maintenance and repairs. Maintenance costs can be categorized as scheduled or unscheduled. Scheduled maintenance activities can include inspecting the turbine, controller, and/or tower; adjusting blades; checking production meter and communications components; and providing an overall annual scheduled maintenance visit per the manufacturer's manual. Unscheduled maintenance activities can include a wide variety of activities, ranging from responding to a customer's complaint of noise from the turbine to replacing the generator, electrical components, inverter, blades, or anemometer. Scheduled maintenance site visit costs for a sample of small wind projects were collected for the Benchmarking U.S. Small Wind Costs report (Orrell and Poehlman, 2017). Scheduled maintenance is typically performed annually. That data showed the average scheduled maintenance cost per visit is about $37/kW (32 €/kW); the same value was confirmed by some European companies (Eunice Energy Group, pers. comm.). In general, upon combining different reference sources, it is reasonable to consider O&M cost for small wind projects in the range of 1–3% of the initial investment (Tzen, 2020).

## 3.3 Capacity factors

The economic viability of SWTs depends in a complex way on several factors, including the life-cycle energy production and the possible presence of incentives. To address the first issue, i.e., to correctly evaluate actual production, a key metric is the capacity factor.

Boccard observed mean values below 21% in 2009 (Boccard, 2009), while more recent works observed values between 37% and 40% (Anon, 2015). Figure 6 presents calculated capacity factors for SWTs installed in the United States, based on the average of the first three years of reported generation for each project from the New York State Energy Research and Development Authority and U.S. Department of Agriculture Rural Energy for America Program datasets and the turbine rated capacity (Orrell et al., 2020).

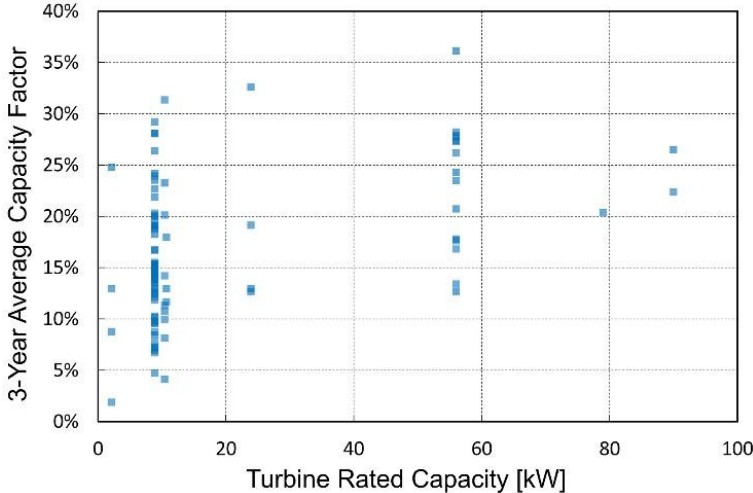


**Figure 6 – Three-year average capacity factor for several U.S. wind projects. Data from (Orrell et al., 2020).**

The three-year average capacity factor for small wind is 17%, but the dataset includes a range from as small as 2% to as high
as 36%. This large variability reflects, more than other variables, the challenges to SWT siting and site suitability. For example,
the capacity factors for the 8.9 kW rated capacity turbines range from 5% to 29%. This means that the same turbine model
sited in different locations can achieve very different capacity factors. Overall, the wind resource quality has the largest impact
on capacity factors, even though technology improvements have raised turbine power outputs significantly. Therefore, the
wide variation of capacity factors across markets is predominantly due to differing wind resource qualities and, to a lesser
extent, the different site configurations and technologies used.
**3.4 Levelized cost of energy**
Scattered data regarding the LCOE of SWTs can be found in literature and relevant reports. One of the most complete databases
is provided by (Orrell et al., 2020), who collected the data reported in Figure 7 (prices are in cents of USD/EUR) for the U.S.
market.



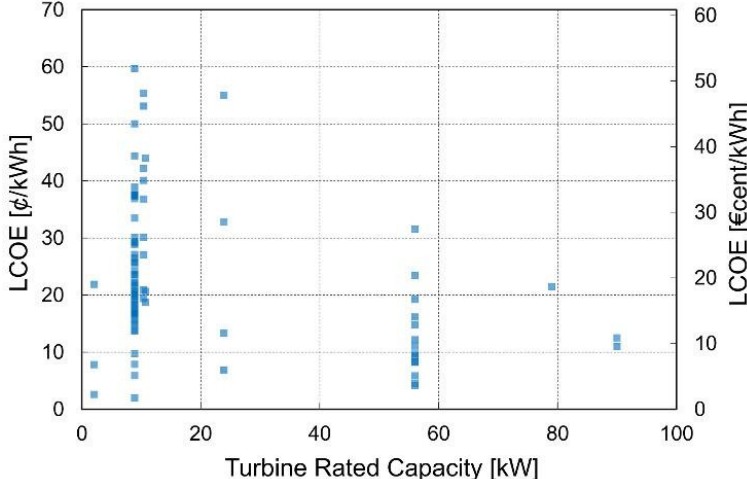


**Figure 7 – Measured LCOE for SWT projects in the U.S. Data from (Orrell et al., 2020).**

The small-wind average LCOE after incentives was 23 ¢/kWh (0.2 €/kWh) (from 86 U.S. projects totalling 2 MW in rated
capacity). To put these numbers in perspective, the LCOE of SWTs may be compared to the average residential retail electric
rates ranging from 8 to 20 ¢/kWh (approx. 7 to 17 €cent/kWh) in the continental United States (Orrell et al. 2019) and to the
LCOE of residential PV, which is below 10 ¢/kWh (8.7 €cent/kWh) (Fu et al., 2018). Recent experiences in Europe for turbines
in the range of 50 to 60 kW showed potential for a significantly lower LCOE on the order of 0.12 €/kWh (0.14 ¢/kWh) (Eunice
Energy Group, pers. comm.). The relationship between calculated LCOEs after incentives and capacity factors is shown in
Figure 8. As expected, the higher the capacity factor, the lower the LCOE in general. Higher capacity factors, which in turn
can reduce LCOEs, can be achieved by better siting, which can help increase energy production and better turbine operations
(i.e., higher turbine availability).





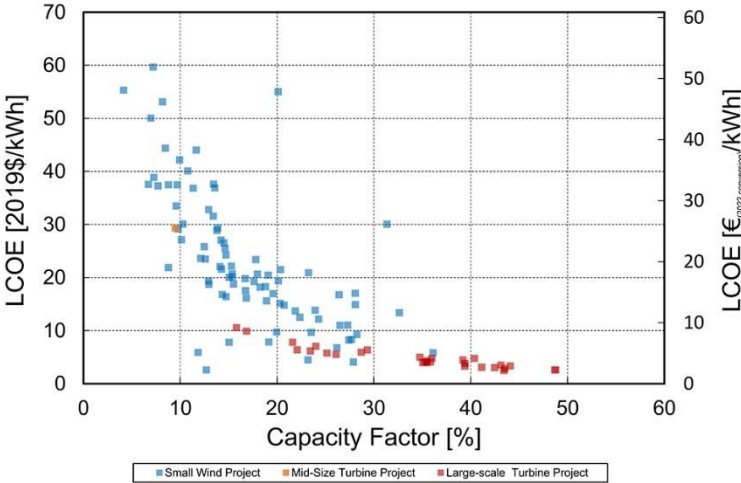

**Figure 8 – Relationship between LCOE and capacity factor for SWT projects. Data from (Orrell et al., 2020).**

Regarding the European Union, to the best of the authors' knowledge, there is no systematic study of the LCOE of SWTs, but there are a number of studies that point to higher LCOEs than those reported for the United States. For a site with a mean annual wind speed of 4.77 m/s, (Bukala et al., 2016) estimate a yearly energy production of 7,551 kWh for an SWT with a rated power of 5 kW, neglecting downtime. They estimate the investment cost of such a wind turbine at 36,500 € (42,000 $), which is lower than that in the data reported for the United States. For a discount factor of 4% and assuming a yearly operation and maintenance cost of 2% of the investment cost, an LCOE of 45 €cent/kWh (52 ¢/kWh) is produced without incentives.

For an SWT with a rated power of 3.5 kW installed at an agricultural site in Belgium with a mean wind speed of 4.13 m/s, (Tordeur, 2018) reports an LCOE of 36 €cent/kWh (41.5 ¢/kWh) without incentives. This, coupled with all the incentives from which an agricultural small-medium enterprise may benefit in Belgium at the time of the measurement campaign (2016) and accounting for a discount rate of 4%, gives a discounted payback time of 19 years. It is worth noting that the true cost of this project was a very low 4,300 €/kW (4950 $/kW). The low cost is partly explained by the fact that the farmer acquired the tower separately at reduced cost and performed most of the installation himself. Even with such major cost-cutting, the SWT is not economically viable, indicating that a mean wind speed of 4.13 m/s is too low for a viable SWT project.

(Bryne, 2017) reports the metered energy output for a number of sites in Ireland. For a site with a mean wind speed of 6.1 m/s, the AEP of a 5.2 kW rated wind turbine is 14,947 kWh, and for a site with a mean wind speed of 4.7 m/s, the AEP of a 2.1 kW rated wind turbine is 3,816 kWh. Assuming again a discount rate of 4%, a yearly operation and maintenance cost of 2% of the investment cost results in LCOEs of 33 €cent/kWh (38 ¢/kWh) and 51 €cent/kWh (59 ¢/kWh) for the 5.2 kW and 2.1 kW turbines, respectively, if the average installed cost per kW from (Orrell et al. 2019) is used. LCOEs of 14 €cent/kWh (16 ¢/kWh) and 22 €cent/kWh (25 ¢/kWh) are produced, respectively, if the average installed cost per kW from (Tordeur, 2018) is used.



Figure 9 presents the results of a study of the LCOE trend versus annual average wind speed at different specific investment
values, with the household energy purchasing prices in EU also shown as references (Predescu, 2016).

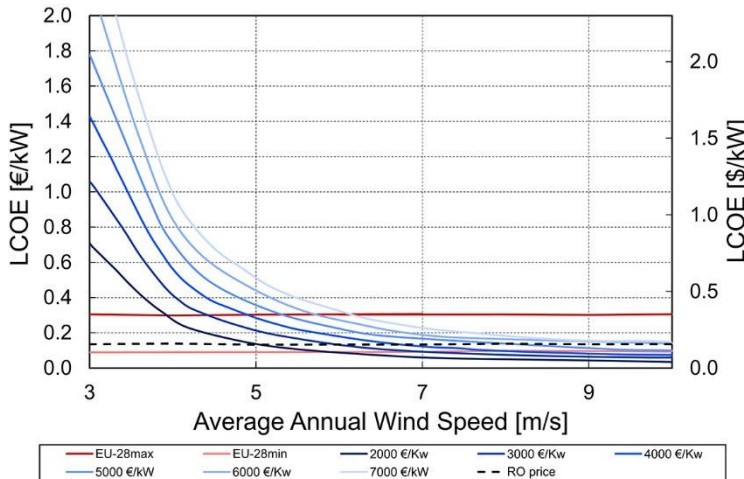

**Figure 9 - LCOE trends versus annual average wind speed at different specific investment values in EU. Data from (Predescu,**
**2016).**

Financial viability for small-wind investment occurs in the region where the LCOE curve, computed for a specific investment
value, is lower than the household energy price at the implementation location. The break-even point for a specific investment
value is at the intersection of the respective LCOE curve with the line representing the household energy price. Beyond this
point toward higher wind speeds, the savings obtained when using small wind technology brings long-term tax-free profit and
savings to the investor. In countries where the household energy price is lower, financial viability can be reached at smaller
specific investment costs and higher annual average wind speeds, which limits the geographical area where grid-connected
small wind systems can be efficient. This analysis shows that in most situations, SWTs cannot compete with residential PV in
terms of economic viability (European Court of Auditors, 2018). Even at sites with high wind speeds, the cost reduction
required to achieve viability is still substantial. Taking the best case from (Bryne, 2017) as a close-to-optimal performance
example with a capacity factor of 33%, the investment cost would need to be less than 6,000 €/kW (6,900 $/kW) for the LCOE
to fall below the 20 €cent/kWh (23 ¢/kWh), which is typical for residential retail electric rates in many European countries.
This illustrates the main conclusion from the above analysis: SWTs may be viable, but only at very windy sites and with a
serious additional effort to reduce the investment cost.



## 4 Status of the technology

While Sections 1–3 reported the status of the technology in terms of diffusion and costs, this section shifts the focus to the specific features of SWTs, which are the core of small wind systems. The philosophy with which this study has been prepared is highlighting those features that make SWTs different from utility-scale machines. This is important for introducing the resulting challenges that must be tackled to further progress SWT technology.

### 4.1 Typical features of small wind turbines compared to utility-scale turbines

Utility-scale wind turbines are usually located in clusters and in areas with high wind resources, from a few turbines to large wind plants located far (e.g., offshore) from the consumer. Although some utility-scale wind turbines may provide energy to the owner, they are typically owned by or provide power to a utility company. In contrast, SWTs are typically owned by the individual or organization that will use the power, such as a home or business, and are installed close to those loads. Because the siting driver for SWTs is proximity to loads and not the optimal wind resource, the winds at these locations often have low average speeds, are highly turbulent, and are more likely to have obstacles nearby, which can create flow structures of a scale commensurable to that of the turbine. On the one hand, this usually leads to lower peak power coefficients, ranging approximately from 0.25 to 0.40 (Wood, 2011), compared to values higher than 0.5 for utility-scale machines (Veers et al., 2019). However, full transparency regarding the real efficiency of SWTs is often missing. For example, in a relatively recent study, it was shown that 15 out of 43 manufacturers claim a power coefficient above the theoretical maximum or Betz–Joukowsky limit (Simic et al., 2013). Notwithstanding this, it is undisputable that the peculiar environment these rotors work in implies that SWTs must be specifically designed to work effectively in both low and turbulent wind resource conditions. The implications of these peculiar working conditions are many and involve all aspects of turbine design and operation, as summarized below.

**Aerodynamics**

The combination of dimensions much smaller than those of utility-scale machines with turbulent winds may present significant problems for the aerodynamics of SWTs. First, the resulting low Reynolds numbers (Re) may cause a laminar separation bubble, which is associated with a local maximum of the drag coefficient in the polar and a reduced lift-to-drag ratio (L/D) (Selig, 2003). The presence of transition and the relative impact of inflow turbulence on it is key for airfoil performance (Abbott and Von Doenhoff, 2010). This has many implications for design, including the fact that airfoils for SWTs must be selected from those that provide good performance at low Re numbers, which favours airfoils with lower thicknesses that are, however, more sensitive to stall. A compromise in this regard must be pursued. The presence of transition makes the L/D dependent on Re and thus is particularly challenging for blade designers. Because the angle needed for maximum L/D is also Re dependent, a constant pitch turbine would not operate at maximum efficiency at a constant tip-speed ratio, making the control strategy in below-rated conditions more complicated (see the following subsection).





The aforementioned issues are particularly challenging in terms of proper simulation. Panel methods usually employed by
companies to define polars likely fail to correctly model these phenomena in many instances, especially in the near- and post-
stall regions. However, accurately modelling these phenomena is crucial for SWTs, particularly stall-controlled ones (Papi et
al., 2021). High-fidelity models used in academia are often not affordable for SWT companies, and airfoil selection is therefore
often based on published performance data. Examples of airfoils with good performance characteristics at low (around $5\cdot10^5$)
Reynolds numbers can be found in (Gigue`re and Selig, 1998; Timmer and van Rooij, 2003). Even high-fidelity turbulence
models, however, often do not predict lift and drag accurately in the presence of transition, let alone laminar separation, and
the designer should rely on lift and drag data measured in reliable wind tunnel tests (Van Treuren, 2015).
The problem of low Reynolds numbers is further exacerbated by the possible installation of SWTs at high altitude
(Pourrajabian et al., 2014), where the air density reduction can substantially reduce Re (up to more than 10%), bringing it to
those values where the effect of transition is more relevant. In this sense, it has been shown that the correction methods
proposed in the standards (wind or power correction) often fail in correctly representing reality.
The influence of blade roughness,  due to insect accumulation in dry areas or leading edge erosion for example, also differs
between SWTs and large turbines. (Holst et al., 2016), for example, discuss the effects of roughness by comparing lift polars
of low-Re airfoils to high-Re utility-scale wind turbine airfoils. Experiments in that study revealed lift deficits of up to 50%
and confirmed the importance of a proper profile selection. In addition, simulations showed that roughness can reduce AEP
by up to 50%. Furthermore, roughness sensitivity could lead to premature separation, especially near the blade root that is
characterized by highly three-dimensional flow (Bangga et al., 2017). Thus, employing airfoils with good aerodynamic
characteristics for the specific blade span and expected operational regime is compelling.

**Control**

Large wind turbines have yaw-drive mechanisms to align the rotor to the mean wind direction. Such devices are much more
expensive for SWTs, especially for small rated-power values (10 kW or less): in these applications, some form of free or
passive yaw has been typically used. The most popular options are then a tail fin or the use of a downwind rotor, e.g., SD Wind
(SD Wind Energy, n.d.), Skystream (XZERES Wind Turbines, n.d.), Carter Wind (Carter Wind Energy, n.d.), and others. The
downwind configuration solution is experiencing a revival for some specific applications in utility-scale machines, especially
for floating offshore applications (Bortolotti et al., 2021). For larger turbines, the same yaw-drive technology in use for utility-
scale machines is instead being increasingly applied.
Another control actuation commonly found in large wind turbines is the blade pitch system that can both regulate power and
slow down the rotor for overspeed protection by aerodynamically changing the blades' angle of attack. However, pitch control
is often not available at the scale of SWTs for economic reasons. Designing and manufacturing a fail-safe pitch system within
the physical constraint of a small hub and the capital cost constraints needed to keep an overall low LCOE are one of the
biggest challenges for the SWT industry. The need for a redundant brake mechanism, in fact, translates into either having
independent pitch actuation (as for the utility-scale machines) or an oversized mechanical brake that could bring the rotor to a



stop in the case of grid connection failure and associated runaway rotor. Both options have proven to be prohibitively expensive in the DWT space thus far, and more economical solutions for avoiding overspeed that have been widely adopted include stall regulation and/or rotating the rotor out of the wind direction via a furling mechanism. An attractive option for smaller SWTs is "electromagnetic braking" by shorting the generator output (McMahon et al., 2015). This obviates the need for a mechanical brake. Several current commercial SWTs such as the Bergey XL 15 (Bergey Wind Power, n.d.) use this cost-reducing strategy. Regarding active pitch, however, a recent study (Papi et al., 2021) highlights how the use of advanced pitch-to-feather control strategies can significantly improve the performance of SWTs through more effective power regulation. It is speculated that the aerodynamic power coefficient could be improved significantly to reach Cp ≈ 0.5, which, together with simpler and therefore more accurate aerodynamic modelling performance, could then justify the higher cost of pitch actuation in an SWT. Blade pitch can also help with start-up torque at low wind speeds, whereas a fixed-pitch rotor must rely on its low wind speed and high angle-of-attack performance to overcome the resistive torque of the drivetrain and generator. A quick starting characteristic is crucial for SWTs because they tend to have more start and stop events compared to their larger counterparts due to higher turbulence levels and lower average wind speeds.

Due to the aforementioned technical and economic issues, stall control is still largely used in SWTs. This latter strategy, however, generates peak loads on the blades that are relatively much higher than those seen in utility-scale machines because the pitch cannot be varied in parking conditions. In addition to the lower efficiency in terms of regulation across the functioning range, the stall control strategy inherently introduces difficulties in predicting the aerodynamics of SWTs because three-dimensional flow aspects and unsteady characteristics make the near- and post-stall regions of the polar curves difficult to capture in aerodynamic models, especially in engineering methods (which can be economically used during the design phase). These difficulties are further compounded in the case of passive-yaw configurations. Skewed inflow and dynamic wake physics are still a topic of research in the wind energy community (Ning et al., 2015; Schepers et al., 2021) and in the case of SWTs, given their more dynamic nature (e.g., higher yaw rates, rotational velocities, and passive yaw), introduce further nonlinearities and unsteadiness in the rotor and tail induction fields, rotor aeroelasticity, and overall turbine response.

**Structural design and (scarce) aeroelasticity modelling**

In the field of large wind turbines, the use of aeroelastic simulation tools has been a consolidated practice for years (Bottasso et al., 2006), as well as required for the certification of the machine itself. In the case of SWTs, the common approach up to a few years ago was to build stiff blades characterized by high safety factors in the structural design in order to avoid significant aeroelastic effects. As discussed, however, somewhat larger SWTs (from about 60 kW and up) are now practically equal in complexity to large wind turbines (e.g., they usually have a variable-speed pitch-torque control system, an active yaw control system and, because they often have a single actuation system for the blades, for safety they require mechanical brakes for the emergency stop). In addition, they are often designed for medium-low wind speeds, so the blade is very large (for the 60 kW blades, it is possible to reach 14–15 m). The experience of many authors of this paper, who had the opportunity in the last decade to collaborate with the small or medium enterprises (SMEs) producing these rotors (IEA, 2014), shows that the use of





aeroelastic simulation tools is important to ensure a quality, safe, and economically sustainable project but is still very
uncommon. One of the few aeroelastic analyses of a 5 kW turbine is described by Evans et al. (2018b). The less frequent use
of aeroelastic models in industry is due mainly to a lack of experience of these companies, which very often come from other
industrial fields (e.g., producers of boats or heavy mechanical systems, etc.) where other design tools such as finite element
codes are primarily used. These companies are often not aware of the availability of good aeroelastic tools in the public domain
(e.g., OpenFAST from the National Renewable Energy Laboratory [NREL] (NREL, 2022)). Finally, another limitation to the
use of aeroelastic simulation tools for SWTs is connected to the lack of easy-to-handle post-processing tools. In fact, standards
require the designer to simulate the wind turbine in power production for different wind values and gusts, but also for a variety
of other operating conditions (starting phase, normal and emergency shutdown, transportation, faults, etc.). This results in a
few thousand simulations that must be analysed to extract maximum loading values for the various sub-components of the
wind turbine, including blades, tower, and drive train, but also pitch and yaw, air gap in the generator, supports, bearings,
brake discs, foundation, etc. In turn, these loads, together with fatigue loads and stress range cycles need to be delivered to the
different partner manufacturers. This process therefore requires automated tools and specific skills that are not always available
outside academia or large manufacturers.

### 4.2 Innovative concepts and VAWTs

Whereas conventional horizontal-axis wind turbines (HAWTs) have become the reference technology for all scales up to 15+
MW, alternative concepts are still being proposed for SWTs (Damota et al., 2015).
A popular modification to small HAWTs is to enclose the rotor with a diffuser to induce more air flow through the blades and
thereby increase the power output. This produces a diffuser-augmented wind turbine (DAWT), some examples of which are
shown in the first row of Figure 10. Adding a diffuser is indeed more attractive for small turbines than large ones, because the
additional structural and wind loads on the latter are likely to be excessive. A diffuser is a relatively simple modification to
basic turbine design, but it is still not clear how to optimize the diffuser and rotor to extract maximum power and whether the
extra power is worth the cost of the diffuser. An interesting review demonstrating the enduring fascination of the concept has
been recently reported by (Bontempo and Manna, 2020). There are other advantages of DAWTs: the diffuser may contain a
blade if it detaches from the rotor, and probably make the turbine quieter and less harmful to birds. These may well be
significant advantages for DAWTs in urban settings (Micallef and van Bussel, 2018).
Beyond other pioneering studies on novel energy-conversion systems such as DAWTs, most of the research on novel SWT
architectures has been directed to vertical-axis wind turbines (VAWTs) (Aslam Bhutta et al., 2012).
Among these, drag-type rotors like the Savonius turbine (Akwa et al., 2012) are relegated to very small applications due to
their low power coefficients and high mass-to-power ratio. Nevertheless, thanks to their simplicity, Savonius VAWTs are still
considered suitable in remote rural areas (e.g., the first electrification of developing countries) (Senthilvel et al., 2020).
On the other hand, despite a long absence from research agendas after the first generation of research culminated in the mid-
1990s, lift-driven VAWTs (or Darrieus concepts) are being increasingly studied (Bianchini et al., 2019). Despite popular





claims, the new understanding of the complex aerodynamics of Darrieus VAWTs achieved in the last decade has proven that
these machines can achieve power coefficients comparable to those of small HAWTs (Bianchini et al., 2015a). More
importantly, VAWTs present several advantages for small-scale applications, namely an intrinsic insensitivity to wind
direction, misaligned flows (Bianchini et al., 2012), or turbulence (Balduzzi et al., 2020), and lower acoustic noise generation
associated with generally lower tip speeds (Möllerström et al., 2016). The advantage of low blade speed, however, is offset by
the need to have a physically bigger, and therefore more expensive, generator and mechanical brake. In addition, VAWTs
allow for a variety of design solutions, which are considered aesthetically pleasant by the public and thus also suitable for
integration in buildings (Dayan, 2006) or with other infrastructure such as streets (Khan et al., 2017). Therefore, a variety of
small manufacturers entered the market either with downscaled VAWTs or with alternative concepts specifically intended for
use on rooftops (Mertens, 2003). Among others, one concept that is receiving increasing attention is the exploitation of the so-
called Magnus effect, which is a phenomenon associated with a solid object spinning in a fluid. This concept has been studied
for both HAWT, e.g., (Sedaghat, 2014), and VAWT designs (Shimizu, 2013). The potential advantage of these solutions lies
in the fact that they can operate in relatively low winds (Bychkov et al., 2007), thus covering a range of winds not typically
exploited by conventional wind turbines.
For very small VAWTs (< 3 kW), recent designs chose high-solidity rotors, i.e., rotors with larger chord-to-radius ratios,
mainly because of the need for sufficiently long chords to increase the aerodynamic forces and the Reynolds number. Based
on recent analyses, this aerodynamic solution seems to provide unprecedented specific power values for small rotors (Bianchini
et al., 2015a). On the other hand, these models showed the significant shortcomings of existing simulation models (Bianchini
et al., 2019), which were resolved largely by the new understanding of the role of flow curvature effects (Bianchini et al.,
2015b, 2016). Renewed research efforts are being undertaken to determine whether VAWTs can fit the scope of distributed
energy production in complex installation areas, as testified to by the recent EU project (Aeolus4Future, n.d.). Parallel to these
research trends, VAWTs are being investigated for deep-water offshore applications with floating substructures (Paulsen et
al., 2013). The more favourable structural loads of the VAWT architecture and the possibility of placing the generator on the
floating platform—and thus lowering the system's centre of mass—may lead to smaller floating supporting structures, better
control, reduced logistics and capital cost, and ultimately a lower LCOE (Arredondo-Galeana and Brennan, 2021). In the realm
of offshore SWTs, floating VAWTs could be deployed in some niche applications like integration with beacons at the entrance
of a port. A recent book, for example, explores the relationships between small wind and hydrokinetic turbines (Clausen et al.,
2021). Overall, despite the benefits that could be provided by VAWTs in some applications, they still lack both theoretical
understanding and technical maturity compared to HAWTs. Whereas the theoretical gap could be overcome by modern
investigation techniques, gaining the same level of industrial maturity as HAWTs seems out of reach at this time. The potential
impact of funded research projects at a national or a broader level could be relevant in proving the real prospects of the
technology and driving their development.
Other touted devices that, at least on paper, have demonstrated the potential for low LCOEs are airborne wind energy (AWE)
kites (Figure 10). They propose to extract wind power either through cross-wind by using lift and therefore flying faster than



the wind speed and carrying turbine generators onboard (fly-gen) or by pulling and unwinding a tether connected to a generator
on the ground (ground-gen). Other concepts expect to take advantage of very high-altitude winds via buoyant aerostat ducts.
None of these concepts has thus far demonstrated an economically viable power curve or has shown successful size scalability
in real-world settings. Yet, there is significant momentum in AWE research, with some pioneering industrial products already
in the market, and the applicability of these devices will likely be in the distributed wind space. While it is difficult to assess
the real costs and LCOE of AWE kites due to their nascent stage, the key advantage they provide is the absence of hefty and
expensive support structures while maintaining a generous rotor swept area. This would have favourable effects on the balance
of station costs that have plagued the DWT industry to date; this is the main reason why they are here mentioned as potential
actors of the small and, more likely, distributed wind market of the future. The challenges these devices face are numerous,
however, from flight safety and reliability to the efficiency of power generation and from the issuing of design and certification
standards to their acceptance by public and aviation authorities, and only future deployments will indicate whether they can
compete in the DWT market.

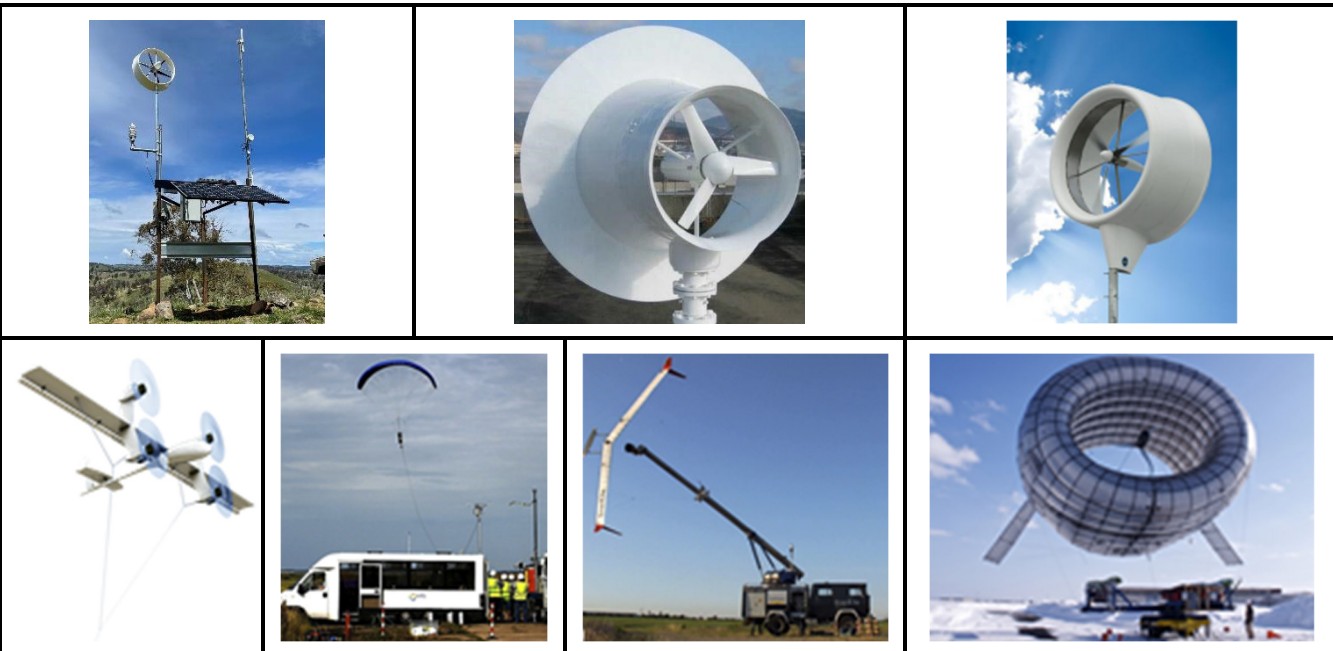

**Figure 10 - Currently proposed DAWT (upper row) and AWE kite archetypes (lower row). First row - from left to right: The Diffuse**
**Energy Hyland 920 diffuser-augmented turbine as part of a remote power system for a communication tower. The 200 W turbine**
**has a maximum diameter of 0.92 m. Photo supplied by Dr Joss Kesby; HAWT with flanged diffuser (Ohya et al., 2008); DonQi**
**urban windmill (photo credit: DonQui Global) | Second row - from left to right cross-wind or fly-Gen (a.k.a. drag-power) devices**
**(image credit: Windlift); ground-gen (a.k.a. lift power) flexible kite (photo credit: KPS); ground-gen rigid kite (photo credit: Ampyx**
**Power); aerostat ducted wind turbine (photo credit: Altaeros).**



### 4.3 Turbine archetypes and design standards

SWTs have not coalesced into a dominant archetype as opposed to the typical utility-scale three-bladed, upwind machines, with many different layouts still being offered in the market. The variety of archetypes (upwind vs. downwind, HAWTs vs. VAWTs, two vs. three or more blades, active pitch vs. stall controlled, etc.; see Figures 11 and 12) creates a challenge for the design standardization and certification of SWTs (Damiani et al., 2022).

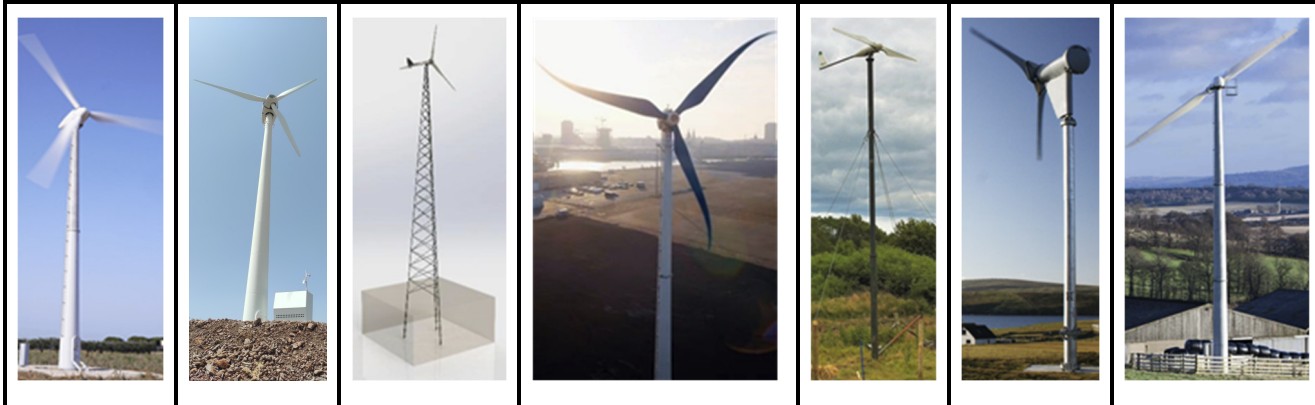

**Figure 11 - Common HAWT archetypes found in the current DWT market. From left to right: Upwind, active pitch and yaw (photo credit: Tozzi Nord); upwind, stall-controlled and active yaw (photo credit: Eunice); upwind, stall-controlled and tailed passive yaw (photo credit: NREL pix 49511); downwind, stall-controlled, passive yaw (photo credit: Eocycle – formerly XANT); upwind, tailed passive yaw, furling (photo credit: Bornay); downwind, pitch or pitch-coning controlled, passive yaw (photo credit: SD Wind [formerly Proven]); downwind, stall-controlled, passive yaw and teeter (photo credit: Ryse Energy [formerly Gaia]).**

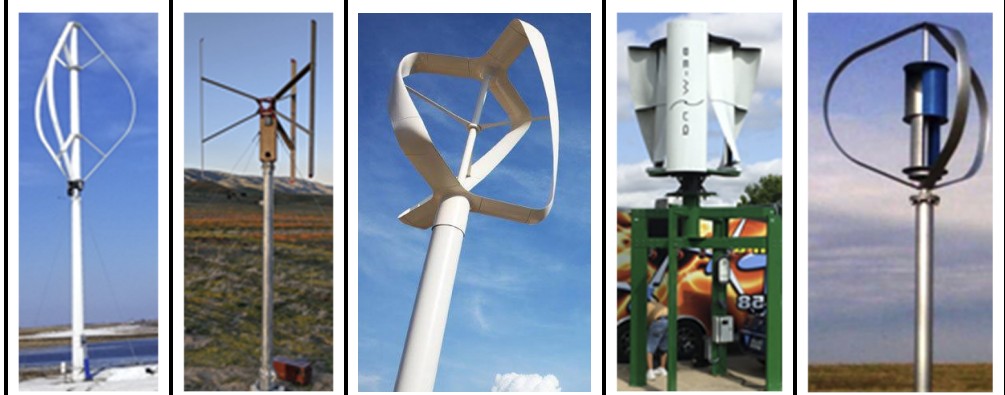

**Figure 12 - Common VAWT archetypes found in the current DWT market. From left to right: Darrieus Troposkien (photo credit: Chava Wind); H-Darrieus (photo credit: Xflow Energy); H-Darrieus with helix shape (photo credit: PRAMAC); Savonius (photo credit: BE Wind); combined Savonius-Darrieus (photo credit: HiVAWT).**






The lack of standardized solutions also indicates that standards for SWT do not have the same level of refinement and
robustness as their counterparts for utility-scale machines.
In the large (onshore) wind turbine industry, the type certification protocol, which primarily follows IEC 61400-1 (IEC:
International Standard, 2019a) for design requirements, entails a detailed design evaluation, manufacturing evaluation, type
testing, and a final evaluation of the product before issuing the type certificate. The design evaluation includes a review of the
aeroelastic modelling used for load determination in all components. These activities can be performed by large, often
multinational, companies with extensive design teams who can afford multidisciplinary development departments, high-
performance computing, and testing facilities. The much smaller companies that manufacture SWTs do not have access to
such resources. For example, even though estimating the loads according to the design standards would require only a few
hours of computational time with state-of-the-art engineering codes, several person-hours are needed to properly set the codes;
the skills of many engineers are often directed more toward mechanical design rather than aero-servo-elastic simulations.
For these reasons, the current design standards differentiate between large and small wind turbines. The IEC standard for wind
turbine design includes a section (Part 2) dedicated to SWTs (IEC: International Standard, 2019b). It covers all mechanical
and electrical subsystems and includes support structure and foundations as well as the grid connection (including power
electronics where applicable). The section applies to wind turbines with a rotor swept area smaller or equal to $200\,\mathrm{m}^2$ generating
at a voltage below 1000 V AC or 1500 V DC and covers both grid-connected turbines and off-grid applications. IEC 61400-2
allows for a number of simplifications to the design and analysis of turbines, including the use of the simplified loads
methodology (SLM) and a reduced number of design load cases (DLCs). The SLM was introduced to provide a straightforward
route to certification in line with the limited resources of SWT manufacturers. However, the SLM has more than doubled the
Safety Factor for Ultimate Loads, which may be easier for the design phase and costs but will create a heavier and more
expensive product/kW. To use the simplified approach, the turbine must have a horizontal axis, two or more cantilevered
blades, coordinated blade movement (not independent and uncoordinated pitching, coning, etc.), and a rigid hub (not teetering
or hinged).
SLM normally leads to a safe but over-designed product. For example, for very small SWTs, the critical DLC includes the
gyroscopic loads on the blade roots and main shaft under yaw. The magnitude of the gyroscopic moment is given by a
simplified load equation involving the blade moment of inertia, the blade angular velocity, and the yaw rate. The SLM safety
factor for this load is 3, although the equation captures in principle the actual physics responsible for the gyroscopic moment
(Wilson et al., 2008). The SLM stipulates the maximum yaw rate as a function of rotor area, and then requires this be multiplied
by the maximum blade angular velocity. The limited information available on SWT yaw behaviour, e.g., (Wright and Wood,
2007) and (Bradney et al., 2019), suggests that high blade speed correlates with low yaw rate, but this is not used in the SLM.
In general terms, our knowledge of the yaw behaviour of SWTs is poor. If the turbine configuration is not covered by the SLM
equations, then alternative simulation modelling or load measurements must be used. Many aspects of the turbine aeroelastic
response that are missed by the SLM approach could, in principle, be captured by higher-fidelity aero-servo-elastic modelling.



Even so, this strategy for the design and certification of SWTs is challenged by the fact that while models are well-tuned for
active yaw and active pitch HAWTs, they are less validated for stall-controlled, passive-yaw HAWTs and progressively less
so for non-traditional archetypes (e.g., teetering hubs, VAWTs, AWE kites) (Damiani et al., 2022). Other aspects of the current
IEC standard and idiosyncrasies related to SWTs are summarized below.
Onshore application sites are classified into classes, as for large turbines, but a special class S is introduced to cover more
extreme conditions, such as those that occur during tropical storms. For the SWT class S, the manufacturer shall describe the
models used and values of essential design parameters in the design documentation. The standard includes the shutdown
procedure for turbines below 40 m², lowering the turbine using a tilt-up tower to bring the rotor out of the wind is permitted,
after which maintenance can be undertaken on the ground. This is realistic and reflects common practice with such very small
turbines. The standard also makes brief reference to off-grid applications that are not simply for battery charging, including
direct connection to electric motors (e.g., for water pumping or even desalination) and heating through directly connected
resistive loads. The guidance on maintenance includes routine inspection of items specific to small turbines, including droop
cables, guy wires, and fasteners.
The reliability of SWTs is guaranteed through duration testing, where at least 6 months of operation is required during which
minimum operation at high winds is stipulated. The standard requires comprehensive documentation of the testing. In addition
to the whole turbine testing, specific component tests are prescribed.
Some SWTs come with design variations. To limit the demands on the original equipment manufacturers, a full design
evaluation is only required on a selected representative configuration. Other variations need only be evaluated or tested in the
ways in which they are different from the representative configuration.
When it comes to power performance testing, IEC 61400-12-1 includes a normative Annex H specifically for the power
performance testing of small turbines. This reflects the fact that testing according to the general standard using 10-minute
averages, where the complete wind speed range must be covered by sufficient data to minimize statistical uncertainty, can be
a time-consuming and expensive process. To get around this difficulty, testing SWTs involves using 1-minute averaged data,
thus considerably reducing the time needed for testing because data points accumulate 10 times as fast, but also because 1-
minute averaging extends the frequency distribution of wind speed, making high-wind-speed data points more common.
The SWT test standard also covers battery charging. Procedures are prescribed that minimize the influence of the specific
battery configuration and condition (state of charge). SWTs that use inverters for grid connection are tested together with the
inverters, and the power measured is the power available to the consumer. Most SWTs lack a clear definition of rated power
and wind speed; instead, a reference power is defined as the averaged power in the 11 m/s bin. Where possible, SWT testing
allows less onerous measurements. For example, battery charging turbines under 40 m² can use less accurate voltage and
current measurement transducers. Wind shear does not need to be measured, and the same applies to relative humidity.
Turbulence correction is not recommended when calculating the power curve.
Comparisons of 10-minute averaged power curves with those based on 1-minute averaged data have been presented in (Elliott
and Infield, 2014). Fortunately, the systematic distortion of power curves due to so-called errors in bins was found to be small.



However, if the 1-minute power curve is used together with a 10-minute averaged wind speed distribution, then an error greater
than 1% in the estimated annual energy yield results. To avoid this, the energy yield calculation should ideally be based on 1-
minute averaged wind speed data. Finally, because the calculation of turbulence intensity depends strongly on the averaging
period, it would be better for this aspect of site characterization to be based on 10-minute data, even if the power curve itself
is based on 1-minute data as prescribed in the SWT test standard.

**5 Grand challenges for small wind turbine technology**

The transition to a more distributed production of energy, combined with the evolution of grids toward "smart" architectures
and control logics, which are more resilient, are leading to an evolution in the way electric services are being provided.
Distributed solar has already demonstrated wide-scale acceptance (IEA, 2019) in this more distributed energy system. While
SWTs have yet to reach general acceptance, they can play a similar and supporting role. To become more commercially
accepted, marked cost and performance improvements are needed. Although significant reductions can be achieved through
understood technology improvements, additional innovations are needed that lie beyond our current knowledge of critical
physics, with particular reference to turbulence, applicability of design assumptions, and the existing modelling and simulation
capabilities. Cost reductions that have been demonstrated within the distributed wind industry show that with adequate
investment, significant hardware cost reductions are possible (US DOE, 2021). However, the generally low investment in
small wind technology research and a lack of consistent and substantial incentive programmes have relegated SWTs to niche
applications with minimal economies of scale. The success of solar PV, which has benefited from significantly more incentive
programmes than SWT in the distributed generation market, demonstrates the importance of stable incentive programmes of
this type in achieving market share.
Among other considerations, a recurring research gap noted in many studies is that SWTs often fail to achieve predicted or
published AEP. This is likely due to a host of considerations such as overly optimistic resource assessments, rotor
underperformance at low wind speeds and during high turbulence, or poor final turbine siting. The two flow features, rotor
underperformance in low winds and/or turbulent winds, are typical of installations on top of short towers and in proximity to
natural or artificial obstacles.
Based on the status of the technology described in the previous sections, the present study identifies five specific grand
challenges (GCs) that must be overcome to spur SWT development and meet the globally expected demand for a wider variety
of distributed energy resources. The grand challenges are visually presented in Figure 13, which represents the graphical
abstract of this study. To address these challenges, a number of unknowns and gaps to be filled are identified (Section 5.1).
Future enablers (Section 5.2) are also suggested as the keys to elevate SWTs to a more mature technology.



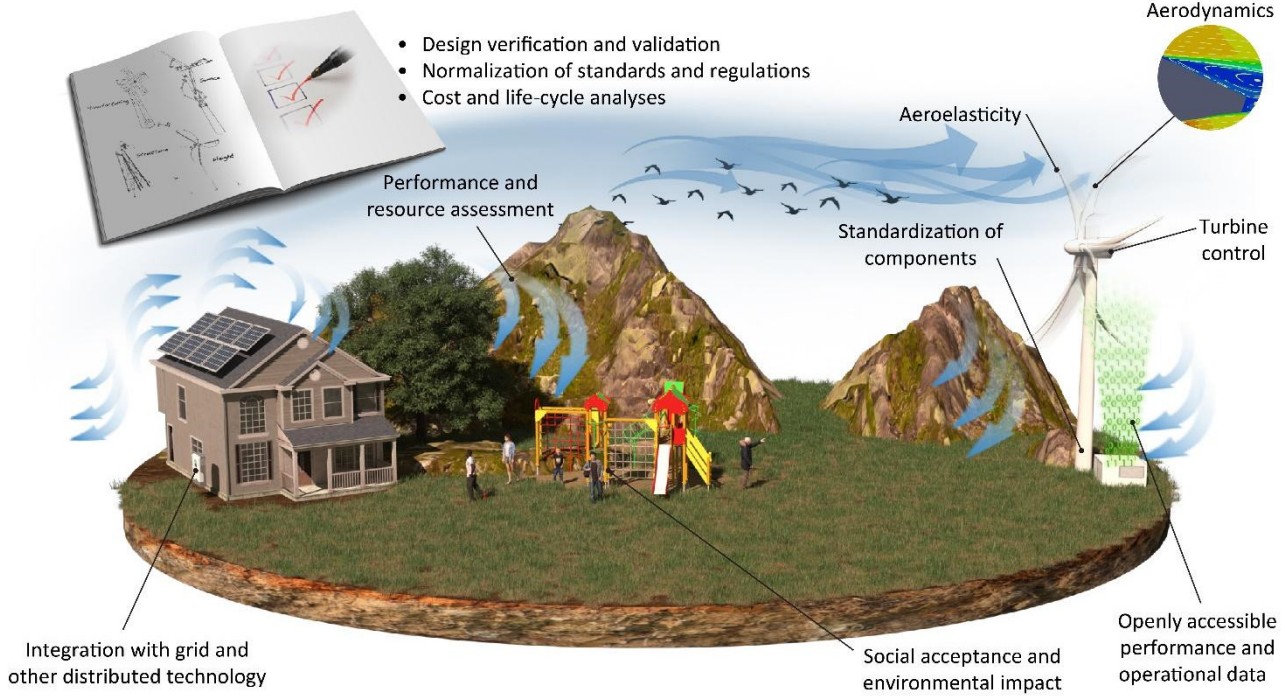


**Figure 13 – Visual synopsis on how the key enablers identified in this study may help tackling the five grand challenges for SWT technology.**



**GRAND CHALLENGE 1 – Improve energy conversion of modern SWTs through better design and control, especially in the case of turbulent wind**

Because SWTs are typically installed in areas with lower (less energetic) and more turbulent wind resources, maximizing the amount of energy that can be harvested from the wind (i.e., maximizing the SWT's capacity factor) while ensuring turbine longevity and survival through infrequent high-wind events is critical. Current wind turbine models have been shown to underperform in comparison to performance based on simulations. This is due to a combination of simulation tools that overpredict turbine performance, driven largely by the simplification of flow features that these turbines are subject to and the actual complexity of the oncoming flow. In particular, better insight into the impact of turbulence and gustiness on turbine performance is needed. This can be achieved with a combination of more detailed testing data and more advanced design tools capable of modelling the complex blade–flow interactions. Additionally, advancements focused to exploit oncoming winds more effectively, including the use of taller towers or the design of lower specific power rotors to better exploit lower winds, must be continued.



To this end, it is now possible to undertake multidimensional blade design to minimize starting time, blade mass, and noise
while maintaining good power extraction and adequate blade strength, e.g., (Sessarego and Wood, 2015). Among other aspects,
blade mass is paramount because it correlates with manufacturing costs and blade inertia. In turn, the ability of a turbine to
start quickly to maximize power extraction at low wind speeds depends on the inertia, as do the gyroscopic loads discussed
above, giving this feature an importance that it does not have for large turbines. SWT blades are naturally stiff and benefit
from additional centrifugal stiffening at high angular speeds, so further optimization should be possible. Because the
gyroscopic loads are major fatigue (as well as ultimate) loads, an improved understanding of turbine yaw behaviour should
allow more optimized turbine design. This should be seen as the key challenge in the modelling of complex unsteady
aerodynamics in the presence of passively yawing rotors, either downwind of the tower or yawed by tail fins.
**GRAND CHALLENGE 2 – Better predict long-term turbine performance with limited resource measurements and**
**prove reliability**
Going beyond accurately optimizing and then predicting the power production of an SWT based on specific wind
characteristics, for SWT projects to receive financing, the industry must be able to accurately predict turbine power production
over the full life of the project. This accuracy of long-term performance prediction is needed to lower the risk associated with
SWTs as seen from the perspective of consumers, insurers, city planning professionals, project financiers, and regulators.
Long-term performance prediction is built on a number of factors, primarily the turbine performance characteristics combined
with accurate wind resource estimation and any changes due to local obstacles over the life of the project. Additionally, turbine
availability due to mechanical, electrical, and weather conditions at the specific site must be considered in addition to long-
term turbine reliability and performance degradation. Although not directly related to turbine design, the availability of spare
and replacement parts, approved turbine repair technicians, company warranty commitments, and specific turbine location
relative to all these factors will also drive long-term power generation.
Beyond corporate credibility of the installer and turbine manufacturer, long-term production reliability can be categorized in
two main areas, i.e., wind-driven resource performance and turbine design reliability. Discussions with the SWT development
community have identified several key challenges to conducting low-cost but accurate resource assessments (Fields et al.,
2016). These include the availability of low-cost anemometer and remote sensing, the lack of high-quality mesoscale modelled
wind speed data at heights typical for SWT installation, and the availability of validated and easy-to-run obstacle modelling to
understand the potential impacts of local obstacles on the wind resource, especially in complex terrain (Duplyakin et al., 2021).
Once an accurate assessment of the resource at the site in question is available, typically for a model year, additional parameters
such as the conditional changes over time, growth of obstructions such as tree cover, and potential weather-driven availability
reduction will need to be added. Tools making resource recommendations must also be verified, providing confidence to
installers, consumers, and the financial community (Tinnesand and Sethuraman, 2019).



Many turbine manufacturers can point to turbines that have operated reliably for many years, but to be successful in today's
market, a long turbine life must be balanced with economic viability (see GC 3). The second element of this challenge is
developing methods that prove SWT technology will operate reliably over the turbine's design life. For example, the SLM of
IEC 61400-2 mandates a simple determination of the total number of fatigue cycles experienced by the blades of an SWT.
Because of the higher angular velocities of SWTs, this is on the order of 100 times the number for large turbine blades. Despite
this, the standard does not mandate fatigue tests for small blades, and there is no strong evidence that fatigue is a major issue
for most SWT blades. On the other hand, the fatigue load case in the SLM appears to be very conservative (Evans et al., 2021).
Addressing this challenge will centre on developing a better understanding of the likely failure modes of SWTs and of the role
of yaw behaviour in gyroscopic fatigue loads, the development and use of validated design tools that address the likely failure
modes, and standards and certification processes to help ensure that turbines operate reliability over their design life. For the
future SWT market to be successful, this effort will need to be accepted by large-scale financial organizations, which are
driving investment in distributed-scale power generations.
**GRAND CHALLENGE 3 – Improve the economic viability of small wind energy**
For an SWT to be economically successful, it must provide reliable power at a cost comparable to other similar technologies,
such as distributed solar PV, and be acceptable by the market. A reduction of the LCOE can be achieved by balancing better
capacity factors (see GC 1) and reducing unit installed cost. Reductions can come from using new materials and manufacturing
techniques, developing standardized solutions for components that can be applied across multiple turbine models, such as
power inverters, and promoting production economies of scale. Moreover, improvements in installation techniques, reducing
the cost of foundations, and other related balance of station costs will be needed.
Many strategies have been considered to lower the cost of turbine hardware, with some solid success in specific turbines. A
balance must however be made to optimize lower turbine costs, which is largely driven by reducing turbine materials and
ensuring successful operation over the turbine's designed life (see GC 2). This optimization must also be balanced with
international standards, which may drive up turbine system costs through the SLM. For example, tools used to predict the
impact of turbulence on component fatigue while load-reducing turbine control, such as adopting pitch regulation typical in
larger rotors, can also help ensure long-term turbine operation while optimizing turbine material needs.
Recent increases in commodity prices as well as supply chain interruptions are causing increased costs for most SWT
manufacturers. Although some of these challenges could be overcome with expanded manufacturing, leading to larger
economies of scale and increased industry purchasing power, expanded research into material substitution for high-cost or
hard-to-access materials would help lower and stabilize turbine manufacturing costs. Expanded work in aligning component
supply across multiple SWT vendors may also help address some high costs and lower component availabilities, especially if
supply chain disruption becomes more common.





Overall, a lower LCOE will also help communities access SWT technology (see GC 5), allowing wind technology to play a
more active role in addressing issues of energy poverty and energy access while reducing the needs for financial incentives,
which typically favour wealthier consumers.

**GRAND CHALLENGE 4 – Facilitate the contribution of SWTs to energy demand and electrical system integration**

Having more distributed wind in the energy mix could contribute significantly to energy justice and power system
decarbonization. Many developing countries, for example, are more likely to have the capacity to build an indigenous SWT
than the solar cells necessary for a PV system. If the fulfilment of GC 2 is pivotal to make investment in SWTs attractive to
many more customers, the introduction of many SWTs to the grid is non-trivial. Although the expanded use of distributed
energy resources will generally require distribution system enhancements, the highly discontinuous power production of
SWTs, which can be hampered by some energy grids with restrictive ramp rates requirements or that are particularly
susceptible to faults, requires additional thinking. SWT technology must not only advance to meet the rapidly evolving grid
code requirements for distributed generation (Preus et al., 2021), but the value they may add to grid reliability and resilience
should be highlighted and monetized. Standardization through improved future revisions of IEC 61400-2 will bring the
industry to a similar technical level for remote control and safety in the smart grids of tomorrow. Due to their distributed
nature, the ability of SWTs to assist load reduction or load shifting in behind-the-meter applications, especially in markets that
are expanding electrification in an effort to reduce carbon production, must be fully assessed and articulated. The ability for
SWTs to complement distributed solar PV technologies will allow improved cost and operability to high renewable
contribution systems for both behind- and in-front-of-the-meter applications (Reiman et al., 2020). The role of energy storage,
and particularly of batteries, will be important not only for wind, but in general for enabling the transition to a smart-user-
based grid paradigm.
The increasing interconnection requirements of all distributed generation, including in many cases two-way communication
with grid control systems, require new SWTs to be more responsive, such as providing low-voltage ride through, more
advanced grid services, and potentially direct grid support. Additionally, with these expanded communication needs, additional
cybersecurity considerations will be required of future SWT technology.
The role of SWTs, however, should not be limited to grid-connected installations. Large global markets for isolated energy
systems, the provision of energy access, and off-grid energy services such as ice making, water pumping, irrigation, or direct
heat could further increase the market potential of the technology and again aid in global decarbonization by offsetting typically
fossil-based means of providing these services.

**GRAND CHALLENGE 5 – Foster engagement, social acceptance, and deployment for global distributed wind markets**

Engaging communities, societies, and regulatory authorities is key for SWT development. Actions need to be taken to enhance
the social understanding of SWTs and to provide evidence that modern turbines are expected to be significantly more efficient





than their predecessors. Turbines must also be designed and deployed while taking into account their installation in proximity
to people and within communities, with a clear understanding of their social and environmental impacts. Expanded research
on community-based impact, such as ice throw and safety setbacks, needs to be carried out, leading to improved standards and
guidelines for turbine installation.
Political and regulatory actions, especially if coordinated among countries on a larger scale, must be enhanced to allow
deployment of the technology in a more effective way. Common regulatory and permitting requirements, based on science and
modern understandings of potential impacts, are needed to streamline development timelines and reduce costs. Incentives,
standards, and promotional policies should also be aligned. This is not only needed in the context of governments, but also
within multi-lateral nongovernmental organizations, development banks, and foundations. For example, the creation of equal
incentives across nations, including a clearly defined timeline for them to stay in place, is needed to encourage investment and
the creation of economies of scale that will be important to sustain each of the other grand challenges.

**5.1 Unknowns and knowledge gaps**

Associated with the grand challenges identified above, the following sections 5.1.1-5.1.6 identify specific focus areas that will
need ongoing consideration if SWT technology is going to be successfully developed to support long-term global needs for
power generation close to loads. In particular, these sections identify the main unknowns and knowledge gaps that need to be
filled to allow the five grand challenges to be accomplished.

**5.1.1 Higher LCOE due to a lack of an economy of scales, keeping the balance of station cost high (in comparison to other renewable-based distributed generation technologies, namely solar PV)**

As discussed previously in the paper, the total global installed cumulative small wind[2] capacity was estimated to be about 1.8
GW as of 2020 (Orrell et al., 2021). In contrast, an estimate of at least 19 GW of residential solar PV was installed in the world
in 2020 alone (IEA, 2020). The difference in installed capacities is driven by a number of factors, including intrinsic siting
requirements, availability of incentives, and differences in costs. In turn, costs are driven by a number of factors as well. In
particular, small wind experiences a lack of economies of scale and high balance of station costs.
Currently, most manufacturing of SWTs is conducted in small plants using batch processes because of the relatively small
manufacturing volume and limited cash from these primarily small companies. Small commercial volumes also increase
component costs, reduce purchasing power, and in times of restricted supply chains, necessitate the ability to substitute
components if traditional ones are unavailable. Each of these items increase cost and complexity and reduce the reliability of
SWT products. As has been clearly demonstrated within the solar industry, large efficiencies and cost reductions can be gained
across the SWT industry by significantly increasing production (Pillai, 2015). To meet diversity and energy justice goals of
the energy transitions, there is a desire for the SWT industry to continue using small plants that are located in the communities

---

[2] This small wind capacity value mostly represents wind turbines up through 100 kW in size, with some capacity from wind turbines up through 250 kW in size.





they are serving rather than transitioning to large manufacturing facilities and thereby the associated shipping and climate
impacts. A transition to serial production, large-volume component purchasing, and advanced manufacturing techniques will
significantly reduce the equipment costs for small turbines.
Balance of station costs include all costs for an installed wind turbine other than the wind turbine and tower equipment. Balance
of station therefore includes costs for customer acquisition; zoning, permitting, inspection, and incentive application;
engineering and design; transportation and logistics; foundation design and installation; electrical infrastructure; turbine and
tower installation and erection; taxes; and overhead and profit. Balance of station costs can represent up to 60% of a small
wind project's total installed cost (Orrell and Poehlman, 2017).
Zoning and permitting costs are considered soft costs and can be particularly burdensome for small wind. For example, at one
point it was reported that potential customers in the Republic of Korea needed written approval from neighbours within a given
radius to install an SWT (Kim, 2018).
Although not typically a direct one-to-one substitution, the lower cost and easier siting of solar PV gives it a competitive
advantage over small wind. From 2008 to 2012, the drop in the overall installed cost of PV systems was mainly due to the
drop in cost of crystalline silicon. Since 2012, installed costs have continued to drop due to decreases in other costs, including
soft costs (Barbose and Darghouth, 2015). In addition, as demand for solar PV increases, production of PV modules can enjoy
the benefit of economies of scale, helping further decrease installed costs.
**5.1.2 Uncertainty in power curves and local wind conditions, resulting in poor estimations of AEP**
The estimation of the AEP of a wind turbine has two main components: the power curve of the wind turbine and the knowledge
of the wind conditions on the site. Nordic Folkecenter's Catalogue of Small Wind Turbines (8th edition) lists 302 types of wind
turbines with a rated power below 50 kW, only a fraction of which have independently measured power curves (Nordic
Folkecenter for Renewable Energy, 2016). This is in stark contrast to large wind turbines, where the vast majority of turbines
have independently measured power curves. If an SWT installation is permitted in the already licensed fields, data from other
wind turbines will help. It is important that the public institutes dedicated for renewable energy sources have calibrated test
sites for this purpose. This would require only one SWT installation for a testing period until it is proven safe and robust. With
this, calibration campaigns could be avoided, and manufacturers and sellers would have lower product costs to compensate
for. This aspect is also partly justified by the costs required to certify the power curve of a wind turbine by an independent
body. These costs are in fact more easily absorbed by large companies involved in the manufacture of multi-MW wind turbines.
Over the past decades, there have been multiple test facilities, some of which are still in operation. While the IEC 61400-2 is
still the most credited reference to standardize performance measurements, some discrepancies still exist with other references
and some aspects are still not completely covered. Further improving this standard could contribute significantly to closing the
gap between small-scale and large-scale wind turbines. Only when a standard is applied to all these aspects will things be
reliable and secure. PVs are more standardized than SWTs, and this is a key for their success.





Because tower heights are commensurate with rotor diameter, SWTs are placed on relatively short towers. Furthermore, tower
heights are often restricted below their optimal values by local planning regulations. Due to wind shear, low towers indicate
lower mean wind speeds and therefore lower production. As discussed, SWTs are strongly affected by installation in altitude,
where the reduction in air density leads to low Reynolds numbers and in turn to a lower aerodynamic efficiency. Furthermore,
the wind flow for SWTs is more likely to be perturbed by nearby obstacles. This has two important effects: (1) the wind pattern
can change over very short distances, making the micrositing of SWTs complex; and (2) the wind is more turbulent. As a
result, even when power curves have been independently measured at a test site, those power curves may not be representative
of real-life performance on the installation site.
The uncertainty in power curves and local wind conditions leads to considerable uncertainty in the estimate of the AEP.
The way to reduce uncertainty in the characterization of local wind conditions is to take on-site wind measurements. However,
site assessment through on-site measurement is often expensive in relation to the installed cost of SWTs and their generation
potential. Deploying instruments for measurement is more expensive and more time consuming than using model-based
approaches to estimate a wind resource, which has led to limited uptake in the use of on-site measurements for small wind
(Tinnesand and Sethuraman, 2019). However, this is not the case for complex terrain, especially because the SWTs are close
to the ground. In these cases, a site assessment is necessary for the project to be bankable.

### 5.1.3 Intermittent incentives and regulations between countries

Incentives applicable to small wind can include net-metering, FITs, other types of production-based payments, grants, rebates,
and tax credits. Regulations that affect small wind can include government renewable energy goals and mandates,
interconnection standards and rules, and utility programmes and rules. Both incentive programmes and regulations vary widely
across different countries. Incentive programmes can vary with respect to the amount and type of funding they provide, what
types of projects are eligible to apply, the cap on the number of projects they support, and the length of time they are available.
Regulations are highly country and utility specific. In countries with complex terrain, good spots are mostly remote (on hills
and mountains rather than in large land fields). Therefore, network expansion from low-voltage to medium-voltage connection
is needed. This increases costs for the investment but simultaneously—and indirectly—helps the distribution companies
expand their network with new equipment.
As discussed in Section 2, Japan, Italy, the United Kingdom, and the Republic of Korea are examples of countries with
intermittent incentive availability and funding levels due to the changes to their FIT programmes over the past approximately
10 years. This changing availability of incentives is one reason why many SWT manufacturers do not remain in the market or
do not participate in certain markets. The fluctuating sales presence of small wind manufacturers both in and exporting from
the United States and China provide examples of how small wind manufacturers must adapt to different market conditions
across countries. In the past, Japan, Italy, and the United Kingdom had been key export markets for SWT manufacturers. With
the programmes discontinued or drastically reduced, the markets are much less attractive, and this contributes to manufacturers
leaving the market. The lack of standardization is another possible reason for manufacturers leaving the market. If there was a



unification (IEC certification, for example), then all the manufacturers could sell globally. For example, six U.S. small wind
manufacturers reported international exports in 2015 with just three in 2020 (Orrell et al., 2021). Similarly, sales in China and
exports from China have fluctuated with the number of Chinese small wind manufacturers in that market. In 2017, only 15
Chinese SWT manufacturers reported sales, a decrease from 28 in 2014 (Duo, 2017), corresponding to a 60% drop in sales
from 2014 to 2017 (Orrell et al., 2021).

### 5.1.4 Lack of openly available data for detailed validation and development of design tools

Aeroelastic modelling should be the primary methodology for structural and performance assessment of any wind turbine.
Such modelling allows the turbine designer to understand and predict the load and power behaviour of the turbine before
witnessing it in the field and to demonstrate the control parameters that have the highest impact on the design and optimize the
configuration most efficiently.
For the results of an aeroelastic model to be used for design and certification, the aeroelastic code (the software), the turbine-
specific inputs, the aeroelastic model setup and usage with those inputs, and the post-processing of the results must achieve a
certain level of verification and validation. Most distributed wind modelers utilize the open-source aeroelastic code
OpenFAST, or the proprietary code HAWC2. While these tools have received adequate validation in past research work, there
remains a need for experimental field data to validate turbine-specific models, especially in the case of SWTs. Publicly
available aeroelastic models are well-tuned for traditional three-bladed HAWTs, although less so for downwind HAWTs, and
are progressively less and less validated for passive yaw, pitch-to-stall, furling, and VAWT machines (Forsyth et al., 2019).
Scarcity of these data is seen in many aspects related to SWTs.
In the validation process, the model results are compared to experimental datasets to ascertain the degree to which the model
represents the actual physics. Therefore, the validation datasets must be properly collected and quality assured. Validation,
however, is not a binary statement about whether a model is valid or invalid, but rather a critical part in the overall assessment
of the suitability of the computational model for the intended application (Hills et al., 2015).
A successful validation exercise requires close collaboration between the experimentalists, the modelers, certification bodies,
and the relevant stakeholders throughout the conceptualization, design, execution, and post-processing phases of the
experiments. Additionally, the computational model should be used to help design the details of the experimental campaign,
which is effectively another (physical) simulation of the true behaviour of the systems.

### 5.1.5 Social acceptance and environmental issues (noise, visual impact, vibrations)

In 2016, some studies suggested that around 70 to 80% of people in Europe support wind farms (Allen, 2016), although there
were still concerns around noise and aesthetics. However, little was known about public attitudes toward SWTs. According to
(Ellis and Ferraro, 2016), the social acceptance of wind energy is influenced by a much wider and complex set of mutual
effects between individuals, communities, place, wind energy operators, regulatory regimes, and technology operating at a
variety of geographical scales. Social acceptance should therefore be viewed within this wider set of relationships and as part





of the transition to a low-carbon economy. In particular, small wind is commonly located closer to customers and therefore
can stimulate social acceptance of wind energy if the installation and the technology used is really adequate and if benefits in
the energy bill are shown.
In 2016, a research survey was completed about the drivers of public attitudes toward SWTs in the UK (Tatchley et al., 2016).
The results showed that half of respondents felt that SWTs were acceptable across a range of settings, with those on road signs
being most accepted and those in hedgerows and gardens being least accepted.
Similar to the results obtained in a survey developed in Europe for the SWIP Project (SWIP Project, 2014) about the awareness
level and public opinion of SWTs, more than 75% of people interviewed showed a positive reaction to the installation of SWTs
in their environment and only 5% showed a negative reaction. Even for all demographic groups involved, the response was
more positive to SWTs than large, utility-scale wind turbines. "Energy Communities" schemes increased this acceptance rate
because more people are able to invest and earn from a wind turbine investment. Generally, people feel detached from large,
utility-scale wind facilities because they do not see the same direct benefits as in the case of SWT investments. Another
conclusion was that industrial sites were regarded as the most acceptable places for installing SWTs, far ahead of the second
place response of roofs in residential areas. Even so, a bad attitude toward SWTs is still noticeable in politics and local
administration in many regions, especially in those countries where historical or aesthetic restrictions are present (e.g., Italy).
In relation to noise emissions, SWT manufacturers have identified noise as a concern (also because some countries do require
noise emission evaluations) and new SWT designs are typically less noisy. However, the general opinion is still that SWTs
are noisy, especially if they are compared with solar PV.
For visual impact (including visual flicker), noise, or safety issues, considerably less concern was shown than toward
performance issues or high investment costs. This is supported by the fact that when an adequate support programme for small
wind is established, social concerns decline. Nevertheless, their visual impact in an urban area can still be a source of concern.
According to (Emblin, 2017), developers must find smart ideas and designs to integrate turbines into communities and to
convince locals that they are the way forward. The visual impact can be kept to a minimum if the turbines are placed carefully
and sensitively. Of course, the design also plays a significant role. These are all issues that may be addressed through
innovations in design and software.
Vibration is another relevant issue, especially in roof-mounted wind turbines with no adequate damping solution and/or SWTs
operating under high-wind conditions regulated by passive power regulation techniques. In those cases, vibration is transmitted
through the pole to the roof or to the ground. When the turbine is sited near dwellings, residents have been known to express
annoyance.
**5.1.6 Reliability of SWTs not always clear. Relatively high cost of certification.**
As discussed, financial incentives in the form of FITs, direct-pay grants, and tax credits help strengthen the global distributed
wind market. Incentive agencies and other industry stakeholders have worked to formulate and implement programme
eligibility requirements to ensure the public funds used in these programmes are directed to successful projects and





embarrassing failures are avoided. One common strategy is to require third-party certification of the wind turbine system
according to national and international standards. The goal of the standards is to provide meaningful criteria upon which to
assess the quality of the engineering that has gone into an SWT and to provide consumers with performance data that will help
them make informed purchasing decisions, e.g., (IEC: International Standard, 2019b). While certification attests that a wind
turbine has been tested and designed according to requirements in the relevant standards, a third party cannot guarantee that a
turbine model will exhibit perfect reliability in the field. Therefore, a level of surveillance must be put in place by the
certification body to monitor and respond to field failures, in collaboration with the turbine manufacturer.
While certification helps improve the reliability of deployed wind turbines, it comes at a significant cost, although efforts have
been made to reduce the complexity and cost of meeting standards for SWTs. To achieve certification, the turbine must be
field tested for power performance, acoustic noise, safety and function, and durability. The turbine designer must also generate
a significant engineering report documenting the calculation of turbine loads, both extreme and fatigue, and the structural
analysis of the major components in the load path. These test and design reports are then evaluated by a third party, usually an
internationally accredited certification body. If the work is found to conform with the applicable standards, certification is
granted, making the turbine model eligible for financial incentives. The validity of the certificate must then be maintained
because design changes and reliability issues may arise.
Other certifications or dedicated studies are required for building permission, including structural engineering of the tower,
the foundation (mostly used in the permission phase), and electrical safety (part of the IEC certification) related to protection
from electrical shock and fire.
While it is very difficult to find publicly available data for field testing and reporting, industrial contacts of the authors in
Europe determined that it costs about 200,000 € (230,000 $) for the complete design assessment of an SWT, while field testing
and reporting alone can cost upwards of 85,000 € (100,000 $) and third-party certification can cost up to about 43,000 €
(50,000 $). Small and medium wind turbine manufacturers in the United States have reported that certification costs, including
fees, direct expenses, and labour time, range from $150,000 (134,000 €) to $500,000 (435,000 €) (Orrell et al., 2020).
**5.2 Improvement areas**
By addressing the five identified grand challenges, SWT technology is expected to decrease significantly in cost, become more
accepted within the distributed energy investment community, and demonstrate acceptable community impact to allow direct
community-based acceptance. To this scope, the following section reviews some main improvement areas where major
research and development is suggested to allow the global SWT market to flourish.
**5.2.1 Changes in turbine design and control**
The task of designing, manufacturing, and installing SWTs has always been challenging. Suppliers of small wind technology
must produce a product that will be deployed in a wide variety of sites around the globe, maintain reliable operation with
minimal maintenance, and be an economically viable choice. For small wind to maintain a competitive stance in the



international distributed clean energy market, future designs must be further optimized, lowering the LCOE. Unlike the process
used largely for current SWT products on the market, future optimized SWT designs will need to utilize validated aero-servo-
elastic modelling as a design tool starting at the concept phase, utilize low-cost, reliable overspeed protection methods, and
incorporate strategies including design for manufacturing, design for certification, and design for installation, all before initial
prototype testing and ideally in the framework of improved and more detailed standards.
While addressing all these changes in detail would require a book rather than a research paper, some key enabling actions are
proposed in the following, clustered together based on the main technical areas.

**Aerodynamics**
Basic wind turbine aerodynamics lead to the statement that a good blade is composed of good airfoils: "good" in the sense of
having a high lift-to-drag ratio. At the low Reynolds numbers of SWTs, this is a major design challenge that has languished
for over 2 decades. Given the developments in Reynolds-Averaged Navier–Stokes (RANS) turbulence and transition models
over that time, a design methodology is becoming available to overcome the limitations of conventional panel methods in use
up to now. In particular, better modelling of the near- and post-stall region of airfoil polars is key not only to improve stall-
controlled machines, but also to get more reliable estimations of loads in a variety of DLCs prescribed by the standards, thus
leading to better prediction of turbine lifetime and possibly enabling lower safety factors. Innovations at the airfoil level should
not only focus on pure aerodynamic performance (in terms of high glide ratio, resistance to stall, and low sensitivity to Re
variations), but also on low noise to make turbines more suitable for installations in proximity to populated areas (certification
labels could also be useful in this regard).
The introduction of *smart blade* technologies for flow control in SWTs may provide a significant boost toward better designs
in the near future. For example, the potential of retrofitting SWTs with passive flow control elements such as vortex generators
and Gurney flaps to improve their starting behaviour and to reduce the risk of stall caused by roughness has recently shown
very promising prospects (Holst et al., 2017).

**Aeroelastic modelling**
Up to now, SWT blades have been much stiffer than large blades and protected by large safety factors in their structural design.
To allow for a blade weight/cost reduction and more efficient designs, aeroelastic modelling should be increasingly used in
SWTs, as it is used for utility-scale turbines. To enable wider use of this simulation tool for design and optimization, gaps and
barriers to its use must be identified and solutions implemented (Damiani et al., 2022). Growth in the theoretical knowledge
owned by SWT-producing companies and a wider availability of easy-to-set, open-source tools will also be required. To
evaluate the impact of the above, (Evans et al., 2018, 2021) investigated blade fatigue by undertaking aeroelastic simulations
of six SWTs up to 50 kW in rated power using OpenFAST (OpenFAST, 2019). Their research shows that the fatigue DLC in
IEC 61400-2 is unduly pessimistic and that more detailed aeroelastic modelling to allow the design of fatigue-resistant blades
at lower cost will be needed. Additionally, the challenge of larger use of aeroelastic models must be supported through





dedicated verification and validation campaigns on a number of different turbine archetypes, sizes, and computational codes.
One particular area of importance for very small turbines is the need for better understanding of yaw behaviour with a tail fin.
Yaw response gives rise to gyroscopic ultimate and fatigue loads, which can be the largest loads on a turbine of around 1 kW
(Wood, 2011). None of the currently available aeroelastic codes contain a tail fin model.
**Control**
Control strategies for SWTs must also evolve to become more robust and cost-effective. We see an example of this evolution
in the contemporary trend of turbine designers moving from tail furling to stall regulation and in some cases pitch regulation.
An example of transition away from furling is the evolution of the Bergey Excel 10 toward the Excel 15 (Bergey Wind Power,
n.d.). The change was in both the increase of power capture via a larger, more efficient rotor and the moving away from the
furling strategy toward a more controlled-stall strategy. Other manufacturers (e.g., Tozzi Nord (Tozzi Nord, n.d.)) are
proposing models with both active yaw and pitch. The difficulty here is to package these controls in relatively tight spaces
while still guaranteeing reliability and redundancy. More research and technical support in this direction is needed because the
experience of utility-scale machines is not directly applicable in SWTs due to cost and physical constraints. However, as
discussed in Section 4, recent studies suggest that the use of pitch control could significantly improve the efficiency of SWTs
(Papi et al., 2021).
**Generator and drivetrain**
The unsteady behaviour of SWTs, especially during start-up, depends on drivetrain and generator resistance (Vaz et al., 2018).
Typically, the wind speed at which an SWT begins power production as the wind increases in strength is significantly higher
than the speed at which it ceases production as the wind dies away (Wood, 2011). The cut-in wind speed is usually an average
of these two speeds and therefore can give a misleading indication of what wind speed is needed for an SWT to start producing
power. In particular, the cogging torque of permanent magnet generators (PMGs) can be a major impediment to very low wind
speed start-up of small turbines. This problem is exacerbated because the current size of the market means that SWT
manufacturers are typically forced to purchase third-party generators that may not match their blade design, resulting in the
need for higher wind speeds to overcome the cogging torque of the generator. Additionally, because there appears to be few
uses for PMGs in the sub-10 kW capacity, there is little market pressure on generator manufacturers to optimize their designs
for SWT applications. Eventually, SWT manufacturers may design and build their own generators, but turbine sales must
expand greatly to warrant this large investment. The design of turbine-specific generators, optimized with specific blade and
rotor design, would require improved understanding of generators, control systems, and the use of modern additive
manufacturing.





**Design strategies**

Knowing that an SWT must be manufactured, tested, certified, and installed, the pressure is on the designer to incorporate this thinking into the design from the initial concept. Actions in this regard are considered key to reduce both the purchasing and the operating cost and, in turn, to make the LCOE of SWTs more competitive.

*Design for manufacturing* involves the manufacturing process in the design process to avoid future issues in fabrication and assembly. *Design for certification* requires conformity with the relevant design standards early in the design process to avoid future issues found in the design evaluation phase of conformity assessment and turbine certification. Lastly, the SWT must be shipped, installed, and commissioned; therefore, *design for installation* strategies must be considered from the early turbine concept. With this in mind, the complete small wind system, including the foundation, tower, inverter, wiring, disconnects, monitoring, nacelle, access platforms, and rotor, will need to be designed in a way that makes the installation process efficient, well thought-out, innovative, and safe.

**Novel concepts**

While continuously improving existing concepts and archetypes, the recent novel designs discussed in Section 4 like DAWT, Darrieus VAWTs, and mostly AWE still deserve attention and research efforts, since they could represent an important future contribution to distributed power production. Novel turbine concepts, however, are not limited only to the individual turbine performance, but should also include holistic considerations of different elements, from economics to social perspectives, which will be further discussed in subsequent sections.

### 5.2.2 Open data from field experiments

Many, but not all, SWT manufacturers remotely monitor the operation of their turbine fleets. For many smaller turbines, monitoring focuses on electrical parameters that are measured as part of the inverter system, but ongoing measurements of many turbine-specific parameters simply increases the cost and maintenance requirements of turbine systems. Sharing that remote monitoring data is an opportunity for researchers and manufacturers to collaborate on a variety of potential research areas that could expand small wind markets while also helping reduce costs. These areas include isolating and identifying the factors that affect why actual performance differs from predicted performance in real-world conditions and then improving performance prediction tools accordingly, improving wind resource assessment data and models for small wind, calculating actual LCOEs, using the performance data to understand wind's complementarity to solar PV, and enabling wind to complement and communicate with other distributed energy resources in the grid of the future. The inability to predict performance consistently and accurately can negatively affect customer confidence in small wind and access to financing. Increasing investor confidence, reducing perceived risk, and decreasing assessment costs with improved tools and datasets will help small wind achieve large-scale deployment. In this regard, however, it must be clarified that the real "performance" of a wind turbine system is the amount of AEP achievable. As discussed in Section 5.1.2, this actually is driven by variables beyond just turbine technology, including, but not limited to, the project's available wind resource, siting (i.e., tower height, local



obstructions, and other micrositing issues), and turbine availability (i.e., downtime for expected or unexpected maintenance or
grid outages). These variables contribute to why accurately estimating small wind project performance can be challenging. A
better prediction of performance can then be synthetized into the proper combination of good resource estimation coupled with
accurate power performance and then with the guarantee that the turbine will provide that same level of power over its design
life. While the current performance prediction tools generally focus on the first of these questions, which is driven by good
resource assessment, they largely do not address the second part, which is failure analysis. Open data for the verification and
tuning of performance prediction tools will then need not only to cover turbine performance vs. actual wind resource, but also
real production vs. time, fatigue, and failure analyses.
Regarding prediction tools, in particular, special attention should be given to open data to *calibrate and further develop design*
*aero-servo-elastic tools* (see Section 5.2.1) in operating conditions outside of turbine-specific validation that may be needed
as part of turbine certification processes. Having detailed field data that may only be available from heavily instrumented
research-grade turbines in the wind tunnel (e.g., those shared in internationally coordinated programmes like those from the
International Energy Agency (IEA) Wind Technical Collaboration Programme) will foster the development of design tools for
SWTs, enabling the modelers to improve the accuracy of the turbine design tools. Data must also be collected over a wide
range of operating conditions, from the standard steady-state operation to predicting the turbine loads, performance, and
lifetime in actual operating conditions. In this sense, the tools can be validated for scenarios that can be significantly different
from one particular site to another site, e.g., different turbulence levels, anisotropy, wind speed, wind direction, ground
stability, etc. An overview of measurement data collected within IEA projects is given in (Schepers and Schreck, 2019). These
projects also provide examples of how international consensus on sharing data will help the users validate their models while
maintaining any needed confidentiality.
**5.2.3 Improvements in installation, maintenance, and life-cycle analysis**
Over the 10 years from 2010 to 2020, the cost for installing residential-scale solar PV systems in the United States has seen an
approximately 64% reduction in benchmark costs. 42% of these costs have been attributed to installation labour and additional
soft costs, such as siting, permitting, sales tax, and overhead (IEA, 2020). Although a smaller percentage of overall total costs,
significant reductions are seen in structural and electrical hardware costs outside of the inverter and solar module. These
installation costs (the total cost outside of the module and inverter) now make up almost 70% of the total installed cost of a
modern residential-scale solar PV system (Feldman et al., 2021). Limited published data exists for similar balance of station
installation specific costs for small wind ((Orrell et al., 2021) as an example), but a 2017 study of the U.S. distributed wind
market shows that similar costs represent 63% of the cost of residential wind systems (Orrell and Poehlman, 2017), which
indicates that if a cost reduction of a similar magnitude as that demonstrated in the solar industry can be achieved for small
wind, this would represent a 25% reduction in the installed costs of small wind systems.



To date, limited systematic analysis has been undertaken to identify methods to reduce the installation costs of small wind technology. Having more of these studies for different countries and environments is considered a key research area for the evolution of small wind systems.

The SMART Wind Roadmap (DWEA, 2016) identifies a set of potential cost-reduction opportunities based on a consensus-based collaboration of small wind industry members. Most of the focus of this work was in the area of turbine hardware cost reductions, but the report does identify tower, foundation, and turbine erection costs as significant cost drivers for small wind, on par with the costs of the turbine hardware itself. Recent work by industry has focused primarily on reducing the costs of towers, primarily developing self-erecting mono-pole towers that provide lower installation and turbine maintenance costs. Recent efforts to reduce installation costs through the DOE-funded Competitiveness Improvement Project (NREL, 2021) have focused on tower and foundation design, including the use of low or no concrete foundations for SWTs, which can greatly reduce turbine installation timelines and costs. Expanded cost reductions could also be expected in site assessment with the expanded use of modelling tools, simplified installation procedures, and reductions in project acquisition and project permitting, each of which needs to be explored in more detail.

Similarly, a full understanding of O&M costs of DWTs is limited. As introduced in Section 3.2, the most recent U.S. Distributed Wind Market Report (Orrell et al., 2021) provides an estimate of cost of 37 $/kW (32 €/kW) per scheduled maintenance site visit, which is typically required annually. This cost has not seemed to decrease over time. In comparison, O&M expenses on a $(€)/kWh-yr basis for residential-scale solar PV systems has dropped by almost 50% over the last 10 years, again demonstrating strong potential for cost savings (Feldman et al., 2021). Maintenance needs of small turbines cover a range of requirements. Most residential and small commercial turbines are designed to require minimal ongoing maintenance, such as bi-annual inspections and potentially blade reconditioning, depending on the environment. Turbines greater than 50 kW in capacity are assumed to undergo more ongoing maintenance, similar to large wind turbines. Ideas that have been identified to support lower long-term maintenance costs include the expanded use of remote monitoring to understand service needs before maintenance is required and expanded turbine structural modelling to eliminate unplanned maintenance. Systematic approaches to reduce maintenance for the distributed wind fleet should also be pursued. Although individual manufacturers have a good sense of long-term turbine-specific component failure rates, no system-wide assessment has been undertaken to focus research efforts into components that have higher service requirements, such as power electronics. This would also represent a key enabler. Focusing on local and national standards will isolate the SWT manufacturers in the borders of their countries. Unification under a common standard (such as IEC) should be proposed as for PVs. History also shows how the SWT market has failed to follow the large wind turbine and PV pace for growth.

Although stories abound of particular SWTs operating for decades, factual data on the full life-cycle cost and performance of many SWTs is limited, reducing the ability to assess the long-term cost of energy for small wind systems. Additionally, the wide variety of turbines, their almost constant change in design, and limited number of operational small turbines that have undergone a full certification to national and international standards also make it challenging to develop meaningful, information-based estimates of life-cycle cost as has been done with other technologies. To support the better full assessment



of life-cycle costs, NREL developed a cost taxonomy for distributed wind (Forsyth et al., 2017) that has been applied in a small number of cases such as (Orrell and Poehlman, 2017). Most work today focuses on articulating costs based on the installed cost of wind technology, making assumptions on maintenance costs and long-term turbine performance. Estimates of life-cycle costs for SWTs at and below 10 ¢/kWh (8.7 €cent/kWh) are being reported but have not been independently demonstrated or verified. A better estimation of life-cycle costs of SWTs is considered a key enabler. In doing so, of critical concern is an accurate accounting of long-term turbine production. Work has been undertaken in relation to an improved estimation of the site-specific wind resource, a topic that is more complicated due to the higher likelihood of local obstructions. Long-term performance production, which could include consideration of long-term wind turbine availability, turbine performance degradation, and increased impact of obstacles such as vegetation growth, have not been systematically considered to date and would definitely improve these estimations (see also Section 5.2.2).

**5.2.4 Regional appreciation of distributed generation and integration with storage systems**

Although historically used in remote and edge-of-grid applications (Hemeida et al., 2022; Duchaud et al., 2019), the continued decrease in the costs of renewable energy generation and storage technologies, combined with incentive programmes and policies to support local generation, have resulted in a wider acceptance of grid-connected distributed generation. With the advent of lower-cost controls, advanced power electronics, and improved communication systems, the use of more distributed power generation is becoming common. Additionally, new efforts to expand clean energy development, paired with the high costs and typically long project development timelines for transmission development, make the use of distributed generation even more cost-effective as a way to support local power development. Lastly, although it typically requires additional expenses and planning, distributed generation can also be used to support grid resilience when combined with storage and other grid-forming technologies. The bold plans of the European Union as well as many other countries around the world in the direction of e-mobility requires significant infrastructure investments to facilitate the millions of electric vehicle chargers that will be installed. This expansion will, however, put an additional large load on existing low-voltage grid infrastructure that, in most countries, is old and extremely expensive to upgrade. The strain on the low-voltage grid cascades toward the medium-voltage infrastructure, which is also coming much closer to its capacity limits.

Enhancing this development while maintaining a reasonable cost involves simultaneously unloading the low- and medium-voltage grid from some capacity through local energy generation and storage. This is possible when buildings and households in local communities are able to become "net prosumers", meaning that they are simultaneously energy producers and consumers. In the future, these prosumers can serve as active members of the energy system network with the ability to exchange energy and offer stabilizing services to the grid. This is achieved through the integration of renewables with storage in combination with decentralized control. Solar has been the first technology to be successfully combined with storage on a residential or local community level, contributing effectively to the "net prosumer" concept. SWTs have been traditionally very simplistic with respect to their design and control, making their combination with storage more difficult. However, numerous current designs include variable-speed full converter AC/DC/AC turbine concepts and have been successfully





integrated with modern storage technologies. The combination of SWTs with fast-response storage systems allows for the
generation of significant quantities of energy at the low-voltage grid level with a simultaneous grid stabilization capability that
is able to unload capacity in an effective manner from the grid. Similarly, combining wind, solar, and storage in many parts of
the world where wind and solar are not typically coincident, either daily or seasonally, could provide expanded benefits to the
low- and medium-voltage energy distribution network. Actions can also be carried out directly on wind turbine design and
control, e.g., integrating fault ride through technologies.
The biggest challenges for this integration involve the volatile nature of wind turbine operation, which requires a very fast
response from the power electronics and storage technology to maintain constant production levels and allow for fast-response
voltage and frequency regulation. However, building on the distributed generation concept into regional development, the
wider use of distributed wind combined with solar and storage at small scales across a region will reduce the variability
experienced with just single units, providing more reliable and less transient power, likely at a reduced cost and certainly faster
than large-scale transmission system development.
To address the expanded need for energy to remote areas not served by current energy infrastructure across the globe, SWTs
in combination with solar, storage, and advanced load control technology is likely to play an expanding role. Although most
investments within the energy access space currently focus on solar and storage, growing energy needs will make it difficult
and expensive to rely on oversized solar and storage facilities to provide full-time power. The use of SWTs and other renewable
energy devices such as pico-hydro and biomass can provide energy at different times than solar, reducing the cost and space
requirements of large storage systems. The limited civil infrastructure and difficulties in providing the on-site service expertise
that is required for larger wind turbines will make SWT technologies more applicable for these more remote applications.
**5.2.5 Shared programmes of incentives and social actions to improve acceptance**
The majority of renewable energy incentives are targeted at large-scale wind projects and wind farms, where scale is a critical
component in a country's wind energy development success rate (Wolsink, 2013). Social acceptability can also be construed
as commercial acceptance in the case of small wind. Wind energy is naturally more complex to diffuse than other energy
alternatives such as solar panels because it frequently involves infrastructure (foundation, tower, and grid interconnection).
If the economic competitiveness of SWTs can progress significantly as a result of improvements in efficiency, manufacturing,
and siting, then the technology could be sustained in the transitory phase by more coordinated political and regulatory actions
at large scale. For example, a federation like Europe could promote the creation of equal incentives in each member state rather
than leaving them to the single-state energy policy. This could create a common, broader market for SWTs, promoting the
development of an economy of scales. Moreover, different from previous practices, the time framework for these incentives
to stay in place should be clearly assessed to reassure investors and companies and prompt them to bid on the technology. In
this context, networks of research institutions like EAWE in Europe, NAWEA in the United States, or wind energy industrials
like WindEurope can play an important role advising regulatory bodies and politicians.



Social acceptance of SWTs could potentially be improved if the drawback on local ecology such as the habitats of birds,
insects, and other small animals, as well as noise and vibrations, can be minimized. While these concerns are largely debated
in utility-scale machines and a vast literature does exist, the environmental impacts of SWTs are not so well defined as a result
of less scientific research on the topic. Additional studies and projects on the topic would also represent an important enabler
to improve acceptance of small wind.
Finally, it is worth mentioning that the diffusion of small wind technology could also be supported by actions that are somehow
a combination of technical and social aspects. A good example of this is a *virtual net-metering approach* (Hellenic Electricity
Distribution Network Operator S.A., 2021). Under this scheme, consumers could install SWTs away from the consumption
meter and liquidate the energy as a classic net-metering. There is a trend where companies try to get "green electricity" from
their providers or through their own investments to compensate for their footprint (Wang, 2013). This will and should get
amplified in the next few years as companies of all sizes try to become greener. These efforts will boost the sector but also in
a more secure and professional way because this "green point system" will push the wind turbine makers toward real power
curves and better products (Simic et al., 2013). Additionally, a link between this type of investment with ESG [environmental,
social, and governance] policies will boost the market even more due to the comparative advantages of SWTs. For example,
many industrial consumers who have already installed PVs may be eager to increase their green electricity, but they may not
have space available for additional PV.

## 6 Conclusions and key enablers for SWT technology

For SWTs to be widely successful, tomorrow's technology will require a new generation of turbines optimized for complex,
low wind speed locations with high turbulence that can also successfully and reliably operate throughout their design life,
producing the power expected when they were installed. Such turbine designs will require higher-fidelity modelling and
simulation to support lower-order tools for design and optimization of turbine systems in complex installation contexts. These
models will need additional open data for validation and calibration, which are currently very scarce. Also, advancements in
control and materials will be needed to improve the energy capture in gusty flows and to reduce the overall cost. Additionally,
these higher-efficiency and reliable turbines must be paired with accurate performance assessment tools to ensure life-cycle
power production, providing confidence to consumers and financiers alike. Finally, these turbines will be more effectively
integrated with storage systems to achieve higher appreciation of small wind for distributed generation.
To make this scenario possible in the near future, the present study suggests five grand challenges for the small wind
community, on which common and synergic efforts should be devoted. These grand challenges translate into:
(1) improve energy conversion of modern SWTs through better design and control, especially in the case of turbulent wind
(2) better predict long-term turbine performance with limited resource measurements and prove reliability
(3) improve the economic viability of small wind energy



(4) facilitate the contribution of SWTs to the energy demand and electrical system integration
(5) foster engagement, social acceptance, and deployment for global distributed wind markets.
To overcome these challenges, previous sections 5.1.1-5.1.6 of the study have presented the main unknowns and gaps that
must be filled, as well as the main improvement areas in which major research and development actions should be devoted.
As a final product of the work, these areas of focus are synthesized below in 10 key **enablers** that, in the authors' opinion,
more than others would represent the catalysts for a significant development of SWT worldwide.

- **Aeroelasticity for SWTs** – If aeroelasticity has represented the main driver of the size and capacity factor of utility-scale machines, its diffusion to SWTs could also be extremely beneficial. For example, an improved aeroelastic design could contribute to reducing the structural safety factors, in turn enabling a blade weight and cost reduction and more efficient designs. To enable wider use of aero-servo-elastic simulation tools for design and optimization, gaps and barriers still need to be identified and solutions implemented, including growth in the theoretical knowledge owned by SWT-producing companies and wider availability of easy-to-set, open-source tools.

- **Improvement in control strategies –** To achieve more effective and robust control, thus maximizing the energy conversion, a transition away from furling toward more controlled-stall strategies is also seen in very small machines. Moreover, some manufacturers are proposing models with both active yaw and pitch. While the implementation of these controls in SWTs is not straightforward due to the difficulty of packaging them in relatively tight spaces while still guaranteeing reliability and redundancy, recent studies suggest that the use of active pitch and yaw controls could significantly improve the efficiency of future SWTs.

- **Improvement in design, with a focus on the characterization of airfoil aerodynamics at low Re** – Improvements in the design of SWTs will be needed at any level, from the rotor-nacelle assembly (e.g., minimization of drivetrain and generator resistance, with particular reference to the cogging torque) to blades' material and cost, or use of cheaper materials for some of the most expensive components such as the towers. Among others, a key area for improvement is defining (possibly validated with experiments) accurate and reliable airfoil polars with the low Reynolds number range that SWT blades usually work with, remembering their strong sensitivity to air density variations due to installations in altitude for example. Having those data available will produce benefits at different levels, including more effective aerodynamic designs, better prediction of loads, and a more reliable definition of turbine control (especially in stall-controlled machines). Special attention should also be given to aerodynamic noise in view of turbine installation in proximity to populated areas.

- **Open data from both wind tunnel and field experiments** – Open data for verification, validation, and optimization of SWTs are seen as a key enabler for the future evolution of the technology. In particular, thanks to the smaller size of SWTs compared to utility-scale machines, they can be placed at full scale or at low scale in a wind tunnel, meaning that reliable testing can take place in the controlled and known wind tunnel environment. Data collected in these conditions would be of particular use for the evolution and calibration of simulation tools. On the other hand, there is also an urgent



need for different open datasets, i.e., related to field measurements of real turbine performance. These will need to not
only cover turbine performance vs. actual wind resource, but also real production vs. time, fatigue, and failure analyses.
● **More accurate performance and resource assessments** – More accurate assessments of both the real performance of
SWTs and the wind resource are key to improving design, siting, and operation. Regarding performance assessment, a
better quantification of several factors could be beneficial, including the impact of turbulence or the effect of obstacles.
For example, a DOE-funded project plans to include obstacle modelling research results as an add-on feature to wind
resource data for the United States available via an application programming interface. Regarding resource assessment,
high-fidelity Computational Fluid Dynamics (CFD) simulations could provide a significant contribution, even though the
economic convenience of their computational cost must still be proven.
● **Variable validation and verification of SWTs, especially for non-traditional archetypes** – Balancing certification
requirements from a regulatory point of view, which prioritizes design thoroughness, model validation, and public safety,
against requests from the original equipment manufacturers for more streamlined and economical approaches to
certification is difficult. Therefore, there is an immediate need for breaking SWTs into categories for load assessment
and validation requirements that account for both size and archetype. Smaller turbines and more established archetypes
would benefit from less onerous requirements in terms of load assessment and validation, whereas more complicated
machines would require a more in-depth review of the prediction capabilities of the code used for design and load
analysis. Verification and validation guidance in the current design standards is limited, and this is one area that requires
more research and data to increase the diffusion of DWT and SWTs.
● **Standardization** – Standardization at different levels is key for further development of SWT technology. First,
*standardization is needed for components to promote an economy of scale*. In particular, it is suggested that generic
products are designed and produced to achieve economies of scale, in turn enabling reduction of the purchase cost of
SWTs. Examples of this could be the design and production of a generic rotor blade family or lighter and easier-to-install
towers. Similarly, research must be focused on the utilization of lower-cost generators, possibly available on the market
with a standardized design. Standardization would come with non-negligible technical challenges but could represent the
key catalyst for reducing the LCOE in the near future. Moreover, more effective *standardization is needed for regulations*
*and standards*. Regulations for SWT installation among different countries are also largely variable and making those
regulations more uniform through international coordination would represent another pillar toward the creation of a stable
market for the technology. Standards should instead evolve along with the changes in the design and operation of new
machines, with a special focus on aeroelastic design and certification. In particular, we suggest that a major enabler could
be the differentiation of standards as a function of turbine archetype.
● **Detailed studies on cost and life-cycle analysis** – To date, limited systematic analysis has been undertaken to identify
methods to reduce the installation costs of small wind technology. Having more of these studies for different countries
and environments is proposed as a key enabler for the evolution of small wind systems, in connection with the impulse
toward standardization. The same applies to life-cycle costs, in which a critical concern is accurate accounting for long-



term turbine production; this should include consideration of long-term wind turbine availability, turbine performance
degradation, and increased impact of obstacles, such as vegetation growth, which have not been systematically considered
to date and would definitely improve the estimations.
● **Grid compliance and integration, including storage systems** – To comply with most of the current grid codes as well
as the upcoming grid code modifications, SWTs of larger rated power should probably mostly become variable speed
and make full use of AC/DC/AC converters. Also, new SWT developments will likely make larger use of fault ride
through technologies because they are becoming compulsory for small-scale generating systems. Beyond this, the
combination of SWTs with fast-response storage systems is thought to be key for allowing generation of significant
quantities of energy at the low-voltage-grid level with a simultaneous grid stabilization capability that is able to unload
capacity in an effective manner from the grid. Similarly, combining wind, solar, and storage in many parts of the world
where wind and solar are not typically coincident, either daily or seasonally, could provide expanded benefits to the low-
and medium-voltage energy distribution network and support the establishment of a significant market for small wind
technology.
● **Shared programmes of incentives and new paradigms to support SWT diffusion, with special focus on social**
**acceptance** – Both incentive programmes and regulations have been widely variable across different countries, making
it difficult for producers to stay in the market. More coordinated political and regulatory actions at a large scale should
be fostered in view of the creation of a broader market for SWTs, thus promoting the development of an economy of
scale. Different from previous practices, the time framework for these incentives to stay in place should be clearly
assessed to assure investors and companies and prompt them to bid on the technology. In this context, networks of
research institutions or wind energy industrials could play an important role in advising regulatory bodies and politicians.
All these actions must be coordinated with a better understanding of the environmental impacts of SWTs so that greater
social acceptance can be achieved.
**Acknowledgments**
This study was originally promoted by the Small Wind Turbine Technical Committee of the European Academy of Wind
Energy (EAWE), which is sincerely acknowledged for their support. Experts from around the world have joined the authors'
group in a joint effort to provide a comprehensive view on the topic and a critical analysis on the status of the SWT/DWT
technology. Special thanks are also due to the National Renewable Energy Laboratory (NREL) and Pacific Northwest National
Laboratory for providing access to their most recent data. Moreover, appreciation is shown to Eunice Energy Group for
supporting the study with its industrial experience. The authors would like to acknowledge the members of "IEA Task 41:
Enabling Wind to Contribute to a Distributed Energy Future" for great collaborations and to the numerous experts that are not
authoring the paper but directly contributed via private discussions and friendly reviews; in particular, the assistance provided



by Mr. Vasilis Papatsiros from Eunice Energy Group was greatly appreciated. Finally, Dr. Francesco Papi from Università
degli Studi di Firenze is acknowledged for his contribution in the consolidation of the document and the final editing.

## Financial support

This work was supported in part by the National Renewable Energy Laboratory (preparation of some graphics), the Pacific
Northwest National Laboratory (text editing), the Publications Committee of the European Academy of Wind Energy
(coverage of part of the APC).

## Competing interests

Some authors are members of the editorial board of *Wind Energy Science*. The peer-review process was guided by an
independent editor. The authors have no other competing interests to declare except what is implied by their affiliation.

## Author contribution

All authors were involved in the original draft preparation, review and editing. AB directed the work and was responsible for
much of the introductory, recommendations and summary material. AO was the main author responsible for Section 2 and 3,
with IBG, GE and RD. GB, AC, JIC, RD, CSF, DI, CNN, GP, MR, GS, BS, DW contributed with all their expertise to Section
4. All the authors contributed to Sections 5 and 6. Much material was shared or moved between sections and editing
responsibilities were comprehensive, so Sect. authorship is never exclusive.

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

XZERES Wind Turbines:, n.d.