# Peer review of "Current status and grand challenges for small wind turbine"

_Wind Energy Science, 2022_

## Author Comment (AC1)

Dear Prof. Clausen,

We would like to thank you for your appreciation of the paper and for the interesting suggestion on DAWTs. Based on your comments, additional data, references, and discussion have been added. Thanks to the additional time available, we have also made a throughout revision of the paper, adding more information and new data, when available.

All modifications and responses to the comments have been highlighted in blue-colored text both in this communication and in the revised version of the paper.

We hope that this revised version can be now worth of publication in *Wind Energy Science*.

Best regards,

*Alessandro Bianchini* on behalf of all the authors.

∘∘∘∘∘∘∘∘∘∘∘∘∘∘∘∘

An excellent review article covering "the state of play" of small wind turbines. Diffuser Augmented Wind Turbines (DAWTs) appear to offer significant performance enhancements over bare wind turbines up to about 2 kW rated capacity. I suggest in this article a little more information on DAWTs and the likely impact this will have on their LCOE.

Thank you for this interesting comment. We agree with the Reviewer that interest in, and commercialization of, small diffuser augmented wind turbines has increased recently. Unfortunately, there does not appear to be any detailed LCOE assessment in the open literature, but the following two references give detailed information on DAWT developments and applications. The description of DAWTs in the manuscript has been extended to include these and the new reference by Visser added.

- Evans, S. P., Kesby, J. E., Bradley, J., & Clausen, P. D. (2020). Commercialization of a diffuser augmented wind turbine for distributed generation. In Journal of Physics: Conference Series (Vol. 1452, No. 1, p. 012014). IOP Publishing.
- Visser, K. D. (2022). Real-world development challenges of the Clarkson University 3 meter ducted wind turbine. In Journal of Physics: Conference Series (Vol. 2265, No. 4, p. 042072). IOP Publishing.

---

## Author Comment (AC2)

Dear Reviewer,

We would like to thank you for your qualified observations, which allowed us to fix some important flaws in the way the paper was presented.

Based on your comments, parts of the paper have been rewritten and additional data, references, and discussion have been added. Thanks to the additional time available, we have also made a throughout revision of the paper, adding more information and new data, when available.

All modifications and responses to the comments have been highlighted in blue-colored text both in this communication and in the revised version of the paper.

We hope that this revised version can be now worth of publication in *Wind Energy Science*.

Best regards,

*Alessandro Bianchini* on behalf of all the authors.

ooooooooooooooooo

**General comments**

The paper provides a systematic review of the current SWT technology particularly in relation to their adoption, cost, technical challenges and challenges that currently persist in this sector of the industry. The paper also presents five grand challenges that this industry faces. Finally, key enablers are presented in the conclusion.

The paper abstract is well written and reflects accurately the content of this work. The introduction introduces the content of the paper to the reader. A number of shortcomings have been identified and given under "Specific comments". Other more minor comments are presented under "Editorial comments".

The work is an important contribution to this stuttering industry and promises to provide a foundation for future directions in policy making, research and a guideline to key industry players.

**Specific comments**

Section 2 - Does this data include also urban wind applications? Please specify given that the authors mentioned that they will "not include urban wind specifically…". This question also applies to the contents of Table 1.

Thank you for pointing this out. Indeed, the sentence was misleading as it was written in first draft. We have now clarified that "the authors decided not to include a specific technical analysis of the needs for urban wind". Table 1 and rest of aggregate data do include all types of turbines, as now specified in the paper.

Table 1 – It is recommended to indicate whether FIT was in place for each of the countries, for the years shown. That would summarize the information conveyed in the text.

The comment is right. We have underlined the data referring to countries/years in which FIT were active.

Table 2 – Is there a particular reason why this data is presented in the form of a table rather than a bar chart or line chart? Please consider a graphical representation.

Done, thanks for the suggestion.

Pg 9 – The authors should discuss in a bit more detail costs that might be associated with siting, resource assessment and turbine positioning. As discussed by the author extensively, especially in grand challenge 2, these considerations are sometimes ignored leading to lower capacity factors with an overall result of lower LCOE. Nevertheless, such assessments are done with associated costs. The authors are encouraged to be more critical on these aspects even though for instance in the sourced Table 4 this cost does not seem to feature (unless this is understood to fall under "Engineering"). Admittedly this cost might be difficult to quantify but its something that the authors may want to at least mention.

As the Reviewer correctly noted, micro-siting and wind resource assessment are critical parts of the project development process, but hard to do properly because of the challenges (in achieving a low-cost but accurate resource assessment) described in Grand Challenge 2.

For example, PNNL is doing research to compare initial performance estimates documented at the time of installation with actual recorded production for small wind turbines and trying to identify reasons for why they don't match. The overwhelming majority of the projects we have reviewed have UNDERperformed compared to their initial estimates.

Following the Reviewer's recommendations, we added a paragraph to the paper that describes these issues and hopefully provides a better lead in to GC2.

Section 4.3 Pg 22-25 – This section is somewhat too descriptive and summarises the IEC standard rather than critically assesses its implications on the SWT industry and state of the art. It could also be that this section may be redundant for the objectives that are being sought after in this work. The authors are invited to re-write critically this part of the text or if necessary remove this section altogether. The latter decision would need to be justified to ensure that it doesn't break the flow of the paper.

Based on your comment, we have shortened and critically revised section 4.3. Please note that deletions were not reported in the revised paper not to impact on readability.

Pg 29 Grand Challenge 5 – An interest action that took place some years ago is the RELY COST action http://cost-rely.eu which dealt with the social acceptability of renewable energy systems including small scale wind energy.

Thank you for the suggestion. The project has been cited and referenced in the paper.

Pg 37 "Control" – For some references, no date is available. Please also see comment on references.

Thank you, all references have been revised.

Line 1167, 1168 – Please double check whether the creation of a Europe wide incentive scheme is at all possible and within the operational lines of the EU. This is to avoid a proposal which may go against fundamental EU modus operandi.

The Reviewer is right. The sentence was partially misleading and was rephrased.

Line 1171 and 1172 – This is a very interesting proposal which could provide guidance on a state by state basis to energy authorities.

Thank you for the comment. We also do agree with the Reviewer's view.

Conclusions – It is recommended to put the key enablers into a separate section before the conclusion. The latter should be used as an overarching, bird's eye view of the work rather than setting out new recommendations.

Thank you for the right suggestion that helped improving clarity. We have re-organized the contents as suggested.

Line 1239 – The following paper could be an interesting addition to your reference list in relation to resource assessment:

D.R. Drew, J.F. Barlow, T.T. Cockerill, M.M. Vahdati, The importance of accurate wind resource assessment for evaluating the economic viability of small wind turbines, Renewable Energy, Volume 77, 2015, Pages 493-500, ISSN 0960-1481, https://doi.org/10.1016/j.renene.2014.12.032.

Thank you again for the interesting reading suggestions. The paper has been studied and added to the references list.

References – Please review all references as there are many of these that are missing basic information. For example Line 1321, line 1324, line 1325 and various others.

References have been checked and corrected. Thank you for pointing this out.

**Editorial comments**

Line 56 – "which constitute" not "which each…"

Fixed.

Line 414 – Cp - use subscript appropriately

Noted and corrected.

Line 814 – "by installation in high altitude" should be "by installation at high altitude"

Corrected.

---

## Author Comment (AC3)

Dear Dr. Sheridan,

Thank you for time you spent reading the paper and for the very interesting suggestions

Based on your comments, parts of the paper have been rewritten and additional data and discussion have been added. Thanks to the additional time available, we have also made a throughout revision of the paper, adding more information and new data, when available.

All modifications and responses to the comments have been highlighted in blue-colored text both in this communication and in the revised version of the paper.

Best regards,

*Alessandro Bianchini* on behalf of all the authors.

ㅇㅇㅇㅇㅇㅇㅇㅇㅇㅇㅇㅇㅇㅇㅇㅇ

The manuscript presents a comprehensive examination of the current worldwide status of small wind turbine design and deployment. Five grand challenges that the small wind community needs to overcome in order to become widely viable, accepted, and competitive are identified and recommendations on how to address such challenges are provided.

The work is timely, well-researched, and needed. I appreciate the comprehensiveness of the paper, focusing on the worldwide small wind market instead of one country or continent and considering details ranging from resource assessment to turbine design to community acceptance.

Thank you very much for your appreciation of the paper.

The following minor considerations are recommended for the final version.

Line 37: Suggest replacing "within" with "for".

Done, thank you.

Figure 1: This graphic is very helpful. I recommend increasing the hub height for the business or community category from "up to 65" as many distributed wind projects feature hub heights of 80 m or even higher.

Changed.

Line 89: It makes sense to exclude urban wind in this analysis, but I am curious as to what thresholds (population, quantity of buildings) you employed to designate urban versus non-urban.

Indeed, there was no direct threshold to designate urban versus non-urban. The sense was that we decided not to address specifically the issues related to wind turbines in a built environments since they are very specific.

Line 184: I am confused by this sentence about trends being stopped or reversed, and wonder if 5 kW is intended instead of 50 kW?

The sentence has been rephrased.

Line 611: It would be helpful to refine "an error greater than 1%". Is the error just a little bit over 1%? Or much higher?

Thank you for pointing this out. The sentence has been rephrased with a more detailed estimation of the error.

Line 666: Suggest rewording to something like "Improve prediction and reliability of long-term turbine performance despite limited resource measurements".
Thanks. We agreed with your suggestion and changed the title of GC#2.

Line 928: In this paragraph, the order of reporting costs in euros and U.S. dollars is inconsistent. It would help the reader if a consistent convention was applied here, with one currency always reported first and the other always reported in parentheses.
Thank you for pointing this out. Notations have been made consistent throughout the paper.